# The cost-effectiveness of preventing, diagnosing, and treating postpartum haemorrhage: A systematic review of economic evaluations

Joshua F. Ginnane [1], Samia Aziz [1,2], Saima Sultana [1,2], Connor Luke Allen [3], Annie McDougall [1], Katherine E. Eddy [1], Nick Scott [1,2], Joshua P. Vogel [1,2]*

1 Maternal, Child and Adolescent Health Program, Burnet Institute, Melbourne, Australia, 2 School of Public Health and Preventive Medicine, Monash University, Melbourne, Australia, 3 Faculty of Medicine, Nursing and Health Sciences, Monash University, Melbourne, Australia

* joshua.vogel@burnet.edu.au

## Abstract

### Background

Postpartum haemorrhage (PPH) is an obstetric emergency. While PPH-related deaths are relatively rare in high-resource settings, PPH continues to be the leading cause of maternal mortality in limited-resource settings. We undertook a systematic review to identify, assess, and synthesise cost-effectiveness evidence on postpartum interventions to prevent, diagnose, or treat PPH.

### Methods and findings

This systematic review was prospectively registered on PROSPERO (CRD42023438424). We searched Medline, Embase, NHS Economic Evaluation Database (NHS EED), EconLit, CINAHL, Emcare, Web of Science, and Global Index Medicus between 22 June 2023 and 11 July 2024 with no date or language limitations. Full economic evaluations of any postpartum intervention for prevention, detection, or management of PPH were eligible. Study screening, data extraction, and quality assessments (using the CHEC-E tool) were undertaken independently by at least 2 reviewers. We developed narrative syntheses of available evidence for each intervention.

From 3,993 citations, 56 studies were included: 33 studies of preventative interventions, 1 study assessed a diagnostic method, 17 studies of treatment interventions, 1 study comparing prevention and treatment, and 4 studies assessed care bundles. Twenty-four studies were conducted in high-income countries, 22 in upper or lower middle-income countries, 3 in low-income countries, and 7 studies involved countries of multiple income levels. Study settings, methods, and findings varied considerably. Interventions with the most consistent findings were the use of tranexamic acid for PPH treatment and using care bundles. In both cases, multiple studies predicted these interventions would either result in better health outcomes and cost savings, or better health outcomes at acceptable costs. Limitations for this

**Data Availability Statement:** All relevant data are within the manuscript and its Supporting information files.

**Funding:** J.F.G. received a Shark Tank Grant from the Burnet Institute, Melbourne (no grant number available) for this review. J.P.V. is supported by an Australian National Health and Medical Research Council Emerging Leadership Investigator Grant (GNT1194248). Funders played no role in the study design, data collection, analysis, narrative synthesis or writing of this review.

**Competing interests:** The authors have declared that no competing interests exist.

**Abbreviations:** CCEMG, Campbell and Cochrane Economics Methods Group; DALY, disability-adjusted life year; GDP, gross domestic product; LMIC, low- and middle-income country; MMR, maternal mortality ratio; NASG, non-pneumatic anti-shock garment; PCVT, point-of-care viscoelastic testing; PPH, postpartum haemorrhage; PPP, purchasing power parity; PSA, probabilistic sensitivity analysis; TBA, traditional birth attendant; UBT, uterine balloon tamponade; VHW, village health worker; WHO, World Health Organization.

review include that no ideal setting was chosen, and therefore, a transferability assessment was not undertaken. In addition, some sources of study uncertainty, such as effectiveness parameters, were interrogated to a greater degree than other sources of uncertainty.

## Conclusions

In this systematic review, we extracted, critically appraised, and summarised the cost-effectiveness evidence from 56 studies across 16 different interventions for the prevention, diagnosis, and treatment of PPH. Both the use of tranexamic acid as part of PPH treatment, and the use of comprehensive PPH bundles for prevention, diagnosis, and treatment have supportive cost-effectiveness evidence across a range of settings. More studies utilizing best practice principles are required to make stronger conclusions on which interventions provide the best value. Several high-priority interventions recommended by World Health Organization (WHO) such as administering additional uterotonics, non-pneumatic anti-shock garment, or uterine balloon tamponade (UBT) for PPH management require robust economic evaluations across high-, middle-, and low-resource settings.

## Author summary

### Why was this study done?

- There are wide array of interventions available to prevent, diagnose, and treat postpartum haemorrhage (PPH).

- The decision on which interventions governments should invest in is challenging, particularly when resources are scarce.

- Systematic reviews on the cost-effectiveness of interventions can assist this decision-making process by providing clear comparisons between options.

### What did the researchers do and find?

- This is, to our knowledge, the first systematic review of economic evaluations covering prevention, diagnosis, and treatment of PPH.

- We identified 56 relevant studies across 16 interventions and summarised the findings from these studies in a comparable way.

- We found consistent evidence that adding tranexamic acid to PPH treatment, and the use of comprehensive care bundles combining preventative, diagnostic, and treatment interventions, are cost-effective.

### What do these findings mean?

- The use of tranexamic acid for PPH treatment has been shown to be either cost-saving or highly cost-effective across multiple settings.

- The combination of multiple interventions into a care bundle is promising—available data suggests these approaches can be cost-effective.

- Sixteen World Health Organization (WHO) recommendations on the prevention, identification of treatment of PPH do not yet have robust cost-effectiveness evidence.

- Study limitations include that no ideal setting was chosen to compare interventions in, and that some sources of study uncertainty were interrogated to a greater extent than others.

## Introduction

Postpartum haemorrhage (PPH) is a time-critical obstetric emergency, defined by the World Health Organization (WHO) as postpartum blood loss of more than 500 ml, regardless of mode of birth [1]. PPH affects approximately 6% of women giving birth and is the most common direct cause of maternal mortality, responsible for an estimated 19.7% of maternal deaths [2,3]. The incidence and resulting maternal mortality caused by PPH are disproportionately concentrated in low- and middle-income countries (LMICs) [4]. In addition to the health considerations, PPH also burdens health systems financially. Births complicated by PPH incur costs 21% to 309% higher than births without complications in LMICs [5]. Since 2015, progress on reducing the global maternal mortality ratio (MMR) appears to have stalled. It seems increasingly unlikely that the Sustainable Development Goals target 3.1—to reduce the global MMR to less than 70 per 100,000 live births by 2030—will be met [4,6]. The 2020 MMR estimates indicate that 223 maternal deaths occur per 100,000 live births worldwide [6]. Renewed efforts are required to address the underlying drivers of preventable pregnancy-related deaths, including the most common cause, PPH.

Deaths from PPH have largely been eliminated in high-income countries, due to coordinated interventions aimed at preventing, detecting, and treating PPH [4]. Preventative interventions include the use of prophylactic uterotonics such as oxytocin, carbetocin, ergometrine/methylergometrine, or misoprostol for all women immediately after birth. Alongside uterotonics, tranexamic acid can also be administered to further reduce the risk of excessive blood loss and the requirement for blood transfusion [7]. To diagnose PPH promptly, assessment of uterine tone and the measurement of blood loss postdelivery is recommended. Blood loss is either visually estimated or measured through collecting blood in a calibrated drape, weighing blood-soaked materials or the use of more sophisticated colorimetric systems [8,9]. If PPH occurs despite preventative measures, treatment should be commenced as soon as possible—this includes uterine massage, additional uterotonics, tranexamic acid, and intravenous fluids. If bleeding continues (i.e., refractory PPH), the use of devices such as uterine balloon tamponade (UBT) or a non-pneumatic antishock garment, compressive measures, and the administration of blood products may be required [10]. More invasive interventions may be required if bleeding continues. These include surgical treatments such as uterine compression sutures, uterine artery ligation or embolization, or hysterectomy [10]. At the health system level, implementing PPH-related care protocols, as well as ensuring adequate training and supervision for health workers, is key to ending PPH-related morbidity and mortality.

Considering the cost-effectiveness of interventions for PPH prevention, detection, and treatment is important for 2 reasons. Firstly, PPH disproportionately affects women in LMICs

—in these countries, financial pressures on the health system mean that difficult decisions are taken around what interventions can be offered [4]. Cost-effectiveness analyses are therefore especially useful to national decision-makers to ensure optimal health impacts for available budget. Secondly, the 2023 WHO Roadmap to Combat Postpartum Haemorrhage has foreshadowed new consolidated WHO PPH guidelines [4]. Developing these guidelines involves explicit consideration of economic evidence, to assess the cost-effectiveness and resource requirements of candidate interventions. Previous systematic reviews on the cost-effectiveness of PPH interventions have solely focussed on the use of uterotonics for PPH prevention [11], or the use of UBT [12] or tranexamic acid for PPH treatment [13]. This systematic review thus aimed to broaden the evidence available from previous reviews by identifying, assessing, and synthesising all available evidence from economic evaluations of any postpartum intervention for PPH prevention, diagnosis, or treatment.

## Methods

For this review, we followed guidelines from the Expert Review of Pharmacoeconomics and Outcomes Research [14] and reported findings in line with the Preferred Reporting Items for Systematic reviews and Meta-analyses (PRISMA) 2020 statement (Table A in S1 Appendix) [15]. We prospectively registered the review protocol on PROSPERO (CRD42023438424). All included articles have been previously published, and ethics approval was not required.

### Eligibility criteria

Full economic studies that have evaluated the cost-effectiveness, cost-utility, or cost-benefit of any method of PPH prevention, diagnosis, or treatment delivered in the postpartum period (from time of birth until 42 days postpartum) were included. We defined full economic evaluations as any evaluation that considered both health consequences and cost consequences of an intervention against a comparator in the single analysis. We considered any such intervention to be eligible, regardless of its effectiveness, or whether it is recommended by WHO or not. Effectiveness studies that included a full economic evaluation such as randomised trials, non-randomised interventional studies, or observational studies (cohort, case-control, cross sectional designs) were also eligible. Partial economic evaluations—those which did not consider both economic and health outcomes compared to a comparator—were excluded. Conference abstracts, protocols, grey literature, and reviews of existing evidence were excluded. There were no limitations based on date or language.

### Information sources, search strategy, and selection process

The search strategy was developed with an expert librarian and run on Medline, Embase, NHS Economic Evaluation Database, EconLit, CINAHL, Emcare, Web of Science, and Global Index Medicus (S2 Appendix). A scoping review of 923 economic evaluations of maternal health interventions was also searched for relevant articles [16]. Resulting citations were uploaded into Covidence software where at least 2 authors (JFG, SA, SS, CA, KE, MDS) independently screened citations by title/abstract level. Full texts of potentially eligible studies were reviewed by 2 reviewers (JFG, SA, SS, CA, KE, MDS). We recorded reasons for exclusion of any full texts. Disputes were resolved through discussion or input from other reviewers.

### Data extraction, synthesis, and quality assessment

Data from each included study was extracted by 2 authors independently into pre-tabulated Excel spreadsheets adapted from previous reviews (Table A in S3 Appendix) [13,16,17]. One

study was only available in Chinese. It was translated for extraction using Google Translate, and the translation confirmed by a native speaking colleague.

Cost-effectiveness analysis is highly dependent on the effectiveness estimate used; the accuracy of these estimates is thus a possible source of bias [18]. For each study, we extracted the effectiveness data used and compared it to an up-to-date effectiveness estimate from the corresponding WHO recommendation and systematic reviews of that intervention. When an included study based its cost-effectiveness calculations on an effectiveness estimate that differed significantly (outside of 95% confidence intervals) from an effectiveness estimate found in the literature, we flagged this as at risk of bias.

Costs were extracted unadjusted, in the year and currency stated by the study. For ease of comparison, all costs were converted to 2023 United States Dollars (USD) using an online tool developed by the Campbell and Cochrane Economics Methods Group (CCEMG) and the Evidence for Policy and Practice Information and Coordinating Centre (EPPI-Centre) [19]. This tool completes a two-stage cost conversion, first adjusting for the cost-year using a gross domestic product (GDP) deflator index, and then a currency conversion using purchasing power parities (PPPs) for GDP [20]. Quality assessments of included studies were completed independently by 2 reviewers (JFG, SA, SS) using the Consensus on Health Economics Criteria extended list (CHEC-E; Table A in S4 Appendix) as this is suitable to evaluate economic evaluations based on clinical trials or modelling studies [14,21,22]. Studies scoring 0% to 49% were considered "low quality," 50% to 74% as "moderate quality," and >75% as "high quality." Disagreements on data extraction or CHEC-E assessment were resolved through discussion or consulting other reviewers.

Studies were grouped for analysis by the characteristics and goal of the intervention, namely PPH prevention, diagnosis, or treatment. Due to heterogeneity of study designs within and between interventions, a narrative synthesis approach was used. A transferability assessment was not completed as no ideal setting was selected to compare to. To identify evidence gaps, we mapped included studies to current WHO recommendations. All guidance from the WHO recommendations for the prevention and treatment of PPH from 2012 were reviewed [1], in addition to the various individual updates that WHO have published to these recommendations up until the end of 2023 [23–29].

## Results

### Characteristics of included studies

Searches identified 3,993 citations, of which, 56 were eligible for inclusion (Fig 1; see Table A in S5 Appendix for excluded studies). Two additional citations were identified after reference review. Of the 58 total eligible citations, 2 (2/58) were found to be additional publications from the same studies meaning only 56 unique studies were identified. Of the included studies, 33 (33/56) assessed PPH prevention interventions, 1 (1/56) assessed prevention versus treatment, 1 (1/56) assessed a diagnostic method alone, 17 (17/56) assessed PPH treatments, and 4 (4/56) assessed combinations (bundles) of prevention and treatment, or diagnosis and treatment (Fig 2). Studies were published between 2006 and 2024 and were conducted in low-income (3/56), lower-middle income (14/56), upper-middle income (8/56), and high-income (24/56) countries. Seven (7/56) studies were completed across multiple income level settings. Thirty-four studies (34/56) were set in hospitals, 13 (13/56) studies assessed interventions across multiple settings, 3 (3/56) studies assessed interventions for home births, 2 (2/56) studies assessed an intervention at a primary health setting, and 4 (4/56) studies did not clearly define which health facilities the studies were set. Of the 56 economic evaluations, 38 (38/56) were models, and 18 (18/56) were part of effectiveness trials. Study characteristics are summarised in

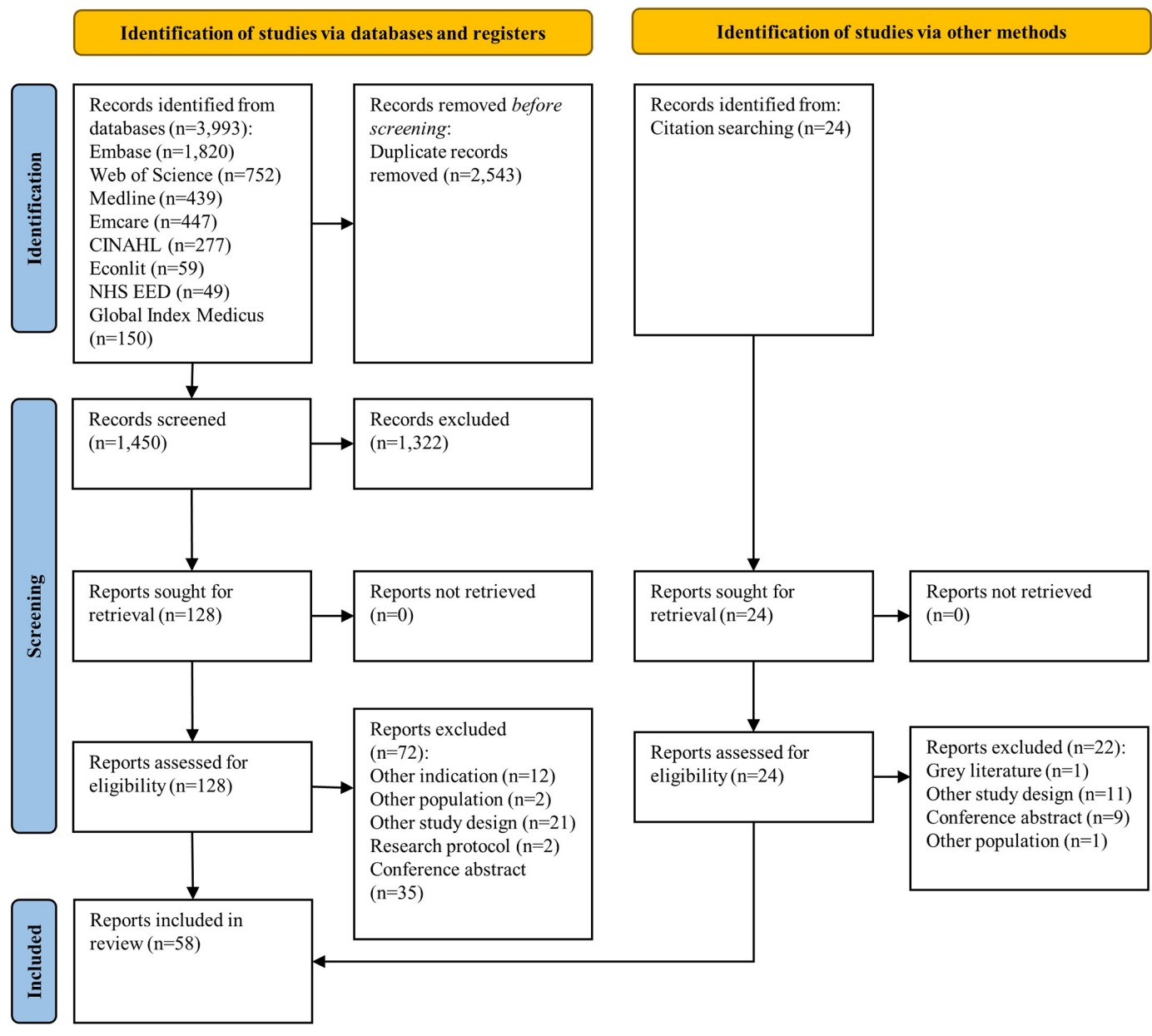

**Fig 1. PRISMA flow diagram.**

Table 1, including the perspective, time horizon, and overall CHEC-E score for each study. Most studies were conducted and reported comprehensively, with 33 (33/56) studies assessed as high quality on CHEC-E, 12 (12/56) moderate, and 11 (11/56) low (detailed study characteristics are presented in Table A in S6 Appendix and full CHEC-E assessments in Tables A–D in S7 Appendix). S8 Appendix provides comparisons of effectiveness data used in each economic analysis, against estimates from published effectiveness reviews. Economic studies on interventions that are recommended by WHO (29 recommendations) were mapped to each recommendation by study country and country income level (S9 Appendix). For 16 (16/29) WHO recommended interventions, we identified zero economic evaluations. For 1 (1/29)

**Postpartum PPH Interventions**

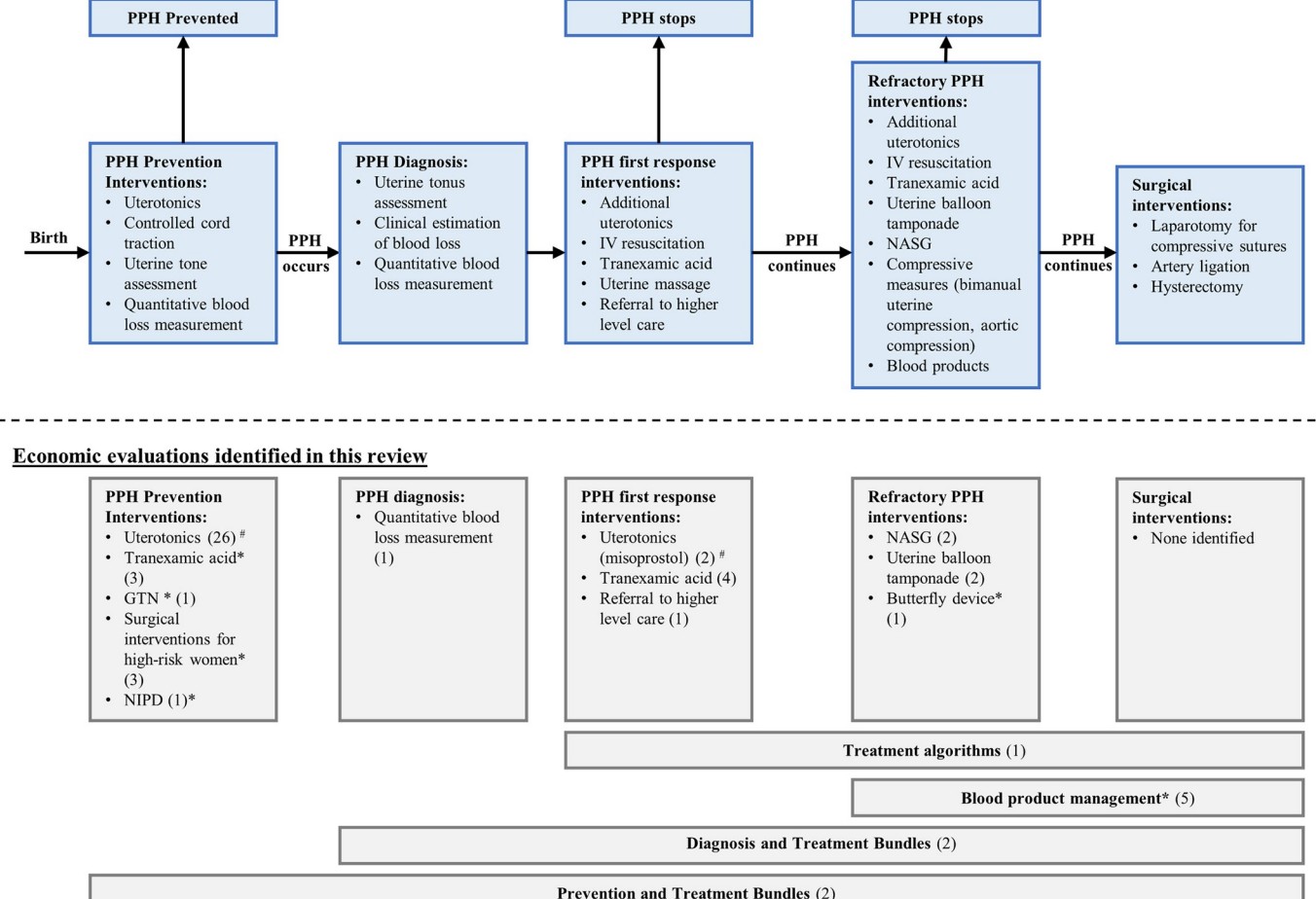

**Economic evaluations identified in this review**

**Fig 2. Interventions and economic evaluations identified.** The top half of this figure represents the timing and order of commonly utilised preventative, diagnostic, and treatment interventions for postpartum haemorrhage. The bottom half of this figure displays the types of interventions assessed in the 56 studies we identified. *Not a WHO recommended intervention. # One study is listed twice as it included assessments of both uterotonics for prevention and uterotonics for PPH treatment. Number of studies identified per intervention type is shown in parenthesise. GTN, glyceryl trinitrate; IV, intravenous; NASG, non-pneumatic anti-shock garment; NIPD, negative intrauterine pressure device; PPH, postpartum haemorrhage; WHO, World Health Organization.

recommendation, we identified studies from high-income countries only, and for 12 (12/29) recommendations, we identified corresponding studies from a range of high-, middle-, or low-income countries.

The results are discussed by intervention type in the text and tables below but can also be viewed by study region and country (S10 Appendix).

## Economic evaluations of preventative interventions

**Prophylactic uterotonics.** Twenty-six (26/56) studies focused on preventative uterotonics (Table 2). A high-quality health technology assessment considered cost-effectiveness of uterotonics for PPH prevention in United Kingdom (UK) hospitals in 2019 [30,31]. For vaginal birth, carbetocin was the most effective—incurring an incremental $1,530.93 per extra PPH > 500 ml averted when switching from oxytocin. For cesarean section, carbetocin was

**Table 1. Summary of included studies.**

| Preventative interventions for PPH | | | | | |
|---|---|---|---|---|---|
| Category | Number of studies | Settings (Country: site) | Comparators considered | Study designs | CHEC-E assessment scores |
| Studies assessing many uterotonics including combinations | Total of 1 study (2 publications) from year:<br>• 2019 [30,31] | • UK: Hospitals [30,31] | • Carbetocin [30,31]<br>• Ergometrine [30,31]<br>• Ergometrine plus Oxytocin [30,31]<br>• Misoprostol plus Oxytocin [30,31]<br>• Misoprostol [30,31]<br>• Oxytocin [30,31] | • Model [30,31] | • High: 1 [30,31]<br>• Moderate: 0<br>• Low: 0 |
| Studies assessing carbetocin as the intervention | Total of 13 studies from years:<br>• 2023 [32]<br>• 2022 [34,35,46]<br>• 2020 [36]<br>• 2019 [40]<br>• 2018 [33,42,45]<br>• 2017 [41,43,44]<br>• 2011 [39] | • Australia: Hospitals [40]<br>• Canada: Hospitals [35]<br>• China: Hospitals [34]<br>• Colombia: Not stated [33]<br>• Ecuador: Hospitals [43]<br>• India: Multiple settings [32]<br>• Malaysia: Hospitals [45]<br>• Peru: Hospitals [42]<br>• Philippines: Hospitals [36]<br>• UK: Hospitals [39,41,44,46] | • Oxytocin [33–36,39–46]<br>• Oxytocin or misoprostol [32] | • EE of effectiveness study [39–41]<br>• Model [32–36,42–46] | • High: 8 [32–34,36,43–46]<br>• Moderate: 3 [35,40,42]<br>• Low: 2 [39,41] |
| Studies assessing oxytocin (various formulations) as the intervention | Total of 5 studies from years:<br>• 2020 [51]<br>• 2016 [47]<br>• 2015 [50]<br>• 2009 [48,49] | • Senegal: Home birth [47]<br>• Peru: Multiple settings [48]<br>• Vietnam: Multiple settings [49]<br>• Latin America and Caribbean countries: Unspecified Health facilities [50]<br>• Bangladesh and Ethiopia: Multiple settings [51] | • Other formulation of oxytocin (e.g., Uniject) [49–51]<br>• Misoprostol [47]<br>• No uterotonics [47–49] | • Model [47–51] | • High: 3 [49–51]<br>• Moderate: 1 [47]<br>• Low: 1 [48] |
| Studies assessing misoprostol as the intervention | Total of 6 studies from years:<br>• 2015 [53,54]<br>• 2010 [52,55,57]<br>• 2009 [56] | • India: Home births [56,57]; Multiple settings [52]<br>• Uganda: Multiple settings [54]<br>• Sub-Saharan Africa: Multiple settings [55]<br>• "International": Multiple settings [53] | • Misoprostol as treatment (rather than prevention) [57]<br>• No uterotonic [52,55–57]<br>• Oxytocin (but only for those with access to a health centre) [53,54] | • Model [52–57] | • High: 2 [52,54]<br>• Moderate: 3 [53,56,57]<br>• Low: 1 [55] |
| Studies assessing AMTSL without specifying uterotonic | Total of 1 study from year:<br>• 2006 [58] | • Guatemala and Zambia: Hospitals [58] | • EMTSL [58] | • Model [58] | • High: 0<br>• Moderate: 1 [58]<br>• Low: 0 |
| Studies assessing tranexamic acid as intervention | Total of 3 studies from years:<br>• 2023 [59,61]<br>• 2021 [60] | • USA: Not stated [59]<br>• France: Hospitals [60,61] | • Standard PPH prevention [59]<br>• Placebo plus standard PPH prevention [60,61] | • EE of an effectiveness study [60,61]<br>• Model [59] | • High: 3 [59–61]<br>• Moderate: 0<br>• Low: 0 |
| Studies assessing GTN as the intervention | Total of 1 study (2 publications) from year:<br>• 2020 [64,65] | • UK: Hospitals [64,65] | • Placebo [64,65] | • EE of effectiveness study [64,65] | • High: 1 [64,65]<br>• Moderate: 0<br>• Low: 0 |
| Studies assessing negative intrauterine pressure suction device | Total of 1 study from year:<br>• 2023 [66] | • India: Hospital [66] | • AMTSL [66] | • EE of an effectiveness study [66] | • High: 0<br>• Moderate: 1 [66]<br>• Low: 0 |

*(Continued)*

**Table 1.** (Continued)

| Studies assessing surgical interventions | Total of 3 studies from years:<br>• 2022 [67]<br>• 2019 [69]<br>• 2017 [68] | • China: Hospitals [67,69]<br>• Italy: Hospitals [68] | • No surgical prevention [67]<br>• Surgical treatment instead of prevention [68]<br>• Normal perioperative care [69] | • EE of an effectiveness study [67–69] | • High: 0<br>• Moderate: 0<br>• Low: 3 [67–69] |
|---|---|---|---|---|---|

**Diagnostic interventions for PPH**

| Study Category | Number of studies | Settings (Country: site) | Comparators considered | Study designs | CHEC-E assessment scores |
|---|---|---|---|---|---|
| Studies assessing quantitative blood loss measurement as the intervention | Total of 1 study from year:<br>• 2020 [70] | • USA: Hospitals [70] | • Visual estimation [70] | • EE of an effectiveness study [70] | • High: 0<br>• Moderate: 0<br>• Low: 1 [70] |

**Treatment interventions for PPH**

| Study Category | Number of studies | Settings (Country: site) | Comparators considered | Study designs | CHEC-E assessment scores |
|---|---|---|---|---|---|
| Studies assessing misoprostol as the intervention | Total of 2 studies from years:<br>• 2010 [57]<br>• 2007 [71] | • Sub-Saharan Africa: Not specified [71]<br>• India: Home birth [57] | • Misoprostol as prevention (rather than treatment) [57]<br>• No medication [57]<br>• Referral to medical centre [71] | • Model [57,71] | • High: 1 [71]<br>• Moderate: 1 [57]<br>• Low: 0 |
| Studies assessing tranexamic acid as the intervention | Total of 4 studies from years:<br>• 2023 [74]<br>• 2022 [72]<br>• 2019 [73]<br>• 2018 [75] | • USA: Hospitals [72,73]<br>• India: Multiple settings [74]<br>• Nigeria and Pakistan: Hospitals [75] | • Standard care [72–74]<br>• Standard care plus placebo [75] | • Model [72–75] | • High: 4 [72–75]<br>• Moderate: 0<br>• Low: 0 |
| Studies assessing NASG as the intervention | Total of 2 studies from years:<br>• 2015 [76]<br>• 2013 [77] | • Zambia and Zimbabwe: Primary health centres [76]<br>• Egypt and Nigeria: Hospitals [77] | • No NASG [77]<br>• NASG only for severe shock [77]<br>• Delayed application of NASG at the hospital [76] | • EE of effectiveness study [76]<br>• Model [77] | • High: 1 [77]<br>• Moderate: 1 [76]<br>• Low: 0 |
| Studies assessing UBTs and similar devices as the intervention | Total of 3 studies from years:<br>• 2023 [80]<br>• 2021 [78]<br>• 2017 [79] | • UK: Hospitals [80]<br>• India: Multiple settings [78]<br>• Kenya: Multiple settings [79] | • Standard care [79,80]<br>• Other UBT models [78] | • Model [78–80] | • High: 2 [78,80]<br>• Moderate: 1 [79]<br>• Low: 0 |
| Studies assessing blood product management interventions including cell salvage | Total of 5 studies from years:<br>• 2018 [82–84]<br>• 2017 [81]<br>• 2014 [85] | • USA: Hospitals [81,82,84]<br>• The Netherlands: Hospitals [85]<br>• UK: Hospitals [83] | • Empiric blood product resuscitation [82]<br>• Alternative blood type and cross matching strategies [81]<br>• Conservative transfusion protocol [85]<br>• No cell salvage [83,84] | • EE of effectiveness study [82,85]<br>• Model [81,83,84] | • High: 4 [81,83–85]<br>• Moderate: 0<br>• Low: 1 [82] |
| Studies assessing treatment algorithms as the intervention | Total of 1 study from year:<br>• 2020 [88] | • Switzerland: Hospitals [88] | • Same hospital, prior to algorithm implemented [88] | • EE of effectiveness study [70,88] | • High: 0<br>• Moderate: 0<br>• Low: 1 [88] |
| Studies assessing inter-facility referral and emergency transport to higher level care | Total of 1 study from year:<br>• 2024 [89] | • Madagascar: Primary Health Centers [89] | • Standard care at primary health centers [89] | • Model [89] | • High: 1 [89]<br>• Moderate: 0<br>• Low: 0 |

(*Continued*)

**Table 1.** (Continued)

| Bundle interventions for PPH | | | | | |
|---|---|---|---|---|---|
| **Study Category** | **Number of studies** | **Settings** (Country: site) | **Comparators considered** | **Study designs** | **CHEC-E assessment scores** |
| Studies assessing prevention and treatment bundles as the intervention | Total of 2 studies from years:<br>• 2023 [92,94] | • Niger: Multiple settings [92]<br>• USA: Hospitals [94] | • Standard care prior to bundle launch [92]<br>• Standard care in non-participating hospitals [94] | • EE of effectiveness study [92]<br>• Model [94] | • High: 1 [94]<br>• Moderate: 0<br>• Low: 1 [92] |
| Studies assessing early detection and treatment bundles as the intervention | Total of 2 studies from years:<br>• 2024 [93]<br>• 2022 [91] | • Wales: Hospitals [91]<br>• Kenya, Nigeria, South Africa, Tanzania: Hospitals [93] | • Standard care prior to bundle launch [91]<br>• Standard care in non-participating hospitals [93] | • EE of effectiveness study [91,93] | • High: 1 [93]<br>• Moderate: 1 [91]<br>• Low: 0 |

AMTSL, active management of the third stage of labour; CHEC-E, Consensus on Health Economics Criteria extended list; EE, economic evaluation; EMTSL, expectant management of the third stage of labour; GTN, glyceryl trinitrate; NASG, non-pneumatic anti-shock garment; PPH, postpartum haemorrhage; TXA, tranexamic acid; UBT, uterine balloon tamponade.

the second most effective and least costly, while misoprostol plus oxytocin was the most effective. This meant that switching from carbetocin to misoprostol plus oxytocin for cesarean section would incur an additional \$4,091.60 per case of PPH $\geq$ 500 ml avoided. Mixed findings across analyses and a degree of uncertainty in the results led the authors to recommend against changing current practice (oxytocin for all births).

Five studies used decision analytic models to compare carbetocin with oxytocin for both vaginal birth and cesarean section [32–36]. Models set in Canadian [35], and Chinese hospitals [34], and Indian primary, secondary, and tertiary health facilities [32] found that implementing carbetocin was dominant (both cost-saving and more efficacious) compared with oxytocin or misoprostol. A Colombian study concluded carbetocin would be either cost saving or cost-effective for cesarean section but not for vaginal birth [33]. A study in the Philippines found carbetocin was not cost-effective for either mode of birth [36]. However, three of these models [33,35,36] used effect estimates that may have either under- [33,36] or over-estimated [33,35] the relative effectiveness of carbetocin [30,37,38] (Table O in S8 Appendix). The 2 studies using effect estimates consistent with current evidence both found carbetocin to be dominant [32,34].

Seven studies, including 3 evaluations from effectiveness studies [39–41] and 4 models [42–45], assessed carbetocin against oxytocin for cesarean section only. Five concluded that carbetocin was the favourable option being either dominant, in the UK [41,44] and Australia [40], or cost-effective in Peru [42] and Ecuador [43]. Two of these studies (including 1 assessed as low quality) [41] may have over-estimated the relative effectiveness of carbetocin [41,42] (Table O in S8 Appendix). A Malaysian study concluded carbetocin was more effective but more costly for cesarean section [45] and did not state a cost-effectiveness threshold to interpret this finding. In contrast, one study concluded that oxytocin was the favourable option for the UK [39], although this study was low quality and used effectiveness data that may underestimate the effectiveness of carbetocin [39] (Table O in S8 Appendix). For vaginal birth, one UK study compared carbetocin and oxytocin in hospitals, concluding carbetocin was likely to be dominant [46]. The study was high quality, and the effectiveness estimates used in the model were consistent with recent systematic reviews.

Five studies assessed oxytocin compared to no uterotonics, non-injectable uterotonics, or other formulations of oxytocin [47–51]. Heterogeneity in their settings, quality and

**Table 2. Results of preventative intervention studies.**

| Study | Intervention/s and comparator/s (dose and route if specified) | Results | Dominance/cost-effectiveness | Summary of study conclusions |
|---|---|---|---|---|
| Pickering and colleagues [31] Note: this analysis was part of a broader HTA—Gallos and colleagues. Only results from analyses of VB are shown in this row, results on CS are shown below in Gallos and colleagues. | 1. Carbetocin 2. Ergometrine 3. Ergometrine plus Oxytocin 4. Misoprostol plus Oxytocin 5. Misoprostol 6. Oxytocin | For VB when considering the costs of SEs: • Carbetocin was the most effective strategy. • Oxytocin was the least costly intervention. • Switching from oxytocin to carbetocin would incur £928 GBP 2016 ($1,530.93 USD 2023) per extra PPH > 500 ml averted or £22,900 ($37,778.41 USD 2023) per severe PPH case averted. • All other options were dominated by carbetocin. | No threshold stated | Current practice in this setting (oxytocin, carbetocin, and ergometrine plus oxytocin) are all favourable strategies based on their relative cost-effectiveness. There was not sufficient evidence to suggest changing current practice at current prices. |
| Gallos and colleagues [30] Note: analyses 1 and 2 (on VB) from this HTA were also published in Pickering and colleagues and are reported in the above row. Only results on CS are presented in this row. | 1. Carbetocin 2. Ergometrine 3. Ergometrine plus Oxytocin 4. Misoprostol plus Oxytocin 5. Misoprostol 6. Oxytocin | For CS, when considering SEs, and excluding ergometrine or ergometrine plus oxytocin (lack of data): • Carbetocin was the least costly intervention and second-most effective strategy. • Misoprostol plus oxytocin is the most effective strategy but is more costly than carbetocin. • All other prevention strategies are dominated by carbetocin, as they are both more costly and less effective than carbetocin. • The estimated ICER for prevention with misoprostol plus oxytocin compared with carbetocin is £2,480.19, GBP 2016 ($4,091.60 USD 2023), per case of PPH of ≥500 ml avoided. • If the impact of SEs were not taken into consideration (analysis 3), then Misoprostol plus oxytocin dominates all other strategies. | Mixed results depending on consideration of side-effects. | The evidence generated in this review were not sufficient to dictate changes to practice in the UK due to the level of uncertainty and mixed results. |
| Barrett and colleagues [35] | 1. Carbetocin as first line prophylactic agent 2. Oxytocin as first line prophylactic agent | In a cohort of 3,242 patients: • 76 PPHs from low-risk VB averted. • 73 PPH from high-risk VB averted. • 154 PPH in CS averted. • Cost savings of $349,000 CAD 2020 ($307,692.81 USD 2023). | Carbetocin dominant compared to oxytocin. | Replacing carbetocin with oxytocin as first-line PPH prevention would reduce costs in this setting. |
| Cook and colleagues [32] | 1. Carbetocin 2. Oxytocin 3. Misoprostol | Base Case 1 • Carbetocin in place of oxytocin reduced PPHs, deaths, DALYs and saved $171,700 USD 2021 ($178,884.78 USD 2023) per 100,000 births. Base Case 2 • Carbetocin in place of misoprostol reduced PPHs, deaths, DALYs and saved $230,248 ($239,882.72 USD 2023) per 100,000 births. OWSA: • Results reported as robust. PSA: • Carbetocin dominant in 98% of iterations against oxytocin and 98.9% against misoprostol. | Carbetocin is a cost-effective intervention and will result in improved health outcomes and lower costs to the Indian public health system. | |

*(Continued)*

**Table 2.** (Continued)

| Study | Intervention/s and comparator/s (dose and route if specified) | Results | Dominance/cost-effectiveness | Summary of study conclusions |
|---|---|---|---|---|
| You and colleagues [34] | 1. Carbetocin (100 μg IV) 2. Oxytocin (10 IU IV bolus) | Base case: <br>• Reduced cost by $29 per birth, USD 2022 ($29.60 USD 2023).<br>• Saved 0.00059 QALY per birth.<br>• Reduced rates of PPH >500 ml, 1,000ml, hysterectomy, and maternal deaths.<br><br>OWSA:<br>• Carbetocin gained more QALYs than oxytocin in every analysis, but expected cost savings were sensitive to changes in relative effectiveness of carbetocin vs. oxytocin.<br><br>PSA:<br>• Carbetocin dominated oxytocin in 99.7% of iterations and was cost-effective in 100%. | Carbetocin dominated oxytocin in 99.7% of iterations. | From the public healthcare perspective in Hong Kong, carbetocin appeared to save total direct medical cost and QALYs for VB or CS. |
| Gil-Rojas and colleagues [33] | 1. Carbetocin (100 μg) 2. Oxytocin (5–10 IU IM for VB or 5IU followed by 30 IU infusion for CS). | CS:<br>• Each PPH prevented saved $94,887 COP 2016 ($86.88 USD 2023).<br>• In 52% of PSA iterations carbetocin dominated and in 15.3% it resulted in increased costs but below the cost-effective threshold.<br>VB:<br>• Carbetocin was the most effective treatment but was more costly.<br>• ICER of $974,790,719 COP 2016 ($1,091,828 USD 2023) per QALY gained.<br>• In only 27% of iterations carbetocin is the dominant alternative and in 7.3% of iterations it is more costly but below the willingness to pay threshold.<br>• In 46.8% of iterations oxytocin remained the dominant alternative. | CS:<br>Carbetocin either dominates or is cost-effective compared to oxytocin (67.3%).<br>VB:<br>Carbetocin not cost-effective at author's stated threshold of $53,090,199 COP 2016 (3× GDP per capita).<br>This is $59,464.47 in USD 2023. | The model for CS showed Carbetocin delivered lower costs and better health outcomes. The model for VB showed carbetocin delivered incremental health gains but at a cost above the specified threshold in Colombia. |
| Briones and colleagues [36] | 1. Carbetocin (100 μg) 2. Oxytocin (10 IU) | Using carbetocin in place of oxytocin resulted in incremental health gains for incremental costs:<br>• CS: $13,187 USD 2019 ($14,317.25 USD 2023) per QALY gained.<br>• VB: $43,164 ($46,863.55 USD 2023) per QALY gained | Carbetocin not cost-effective for either VB or CS at author's stated threshold of $2,895 USD 2019 (1× GDP per capita).<br>This is $3,143.13 in USD 2023.<br>0% chance of cost-effective in VB and 3% chance in CS. | Results suggestive that carbetocin is not cost-effective compared to oxytocin for either VB or CS in the Philippines. |
| Luni and colleagues [41] | 1. Carbetocin 2. Oxytocin | • Significantly lower rate of PPH than the oxytocin group (28% vs. 43%).<br>• No blood products required compared to 17 PRBCs in oxytocin group.<br>• Reduced need for additional uterotonics (7% vs. 48%).<br>• Reduced average medication costs: £10.33 vs. £33.98 GBP, cost year not stated ($17.47 vs. $57.46 USD 2023).<br>• Reduced midwifery costs: £7.69 vs. £52.97 ($13.00 vs. $89.57 USD 2023).<br><br>First year of data collection (2014) used for cost conversion calculation. | Carbetocin dominant compared to oxytocin. | Using carbetocin for PPH prophylaxis in CS reduced the need for blood products, additional uterotonics, and midwifery workload, and has significant potential to provide savings to the delivery suite. |

(*Continued*)

**Table 2.** (*Continued*)

| Study | Intervention/s and comparator/s (dose and route if specified) | Results | Dominance/cost-effectiveness | Summary of study conclusions |
|---|---|---|---|---|
| van der Nelson and colleagues [44] | 1. Carbetocin (100 μg IV)<br>2. Oxytocin (5 IU IV) | Base case:<br>• Carbetocin was less costly, saving £27,518.41 GBP, year not stated ($48,214.13 USD 2023) per cohort of 1,500 deliveries, and more effective with 30 PPH events averted.<br><br>PSA:<br>• Carbetocin dominated in 69.4% of scenarios and was cost-effective in 70.5% of scenarios at author's stated threshold of £20,000 ($35,041 USD 2023) per QALY.<br><br>Year of NHS reference costs from study (2012) used for cost conversion calculation. | Carbetocin dominant compared to oxytocin (in 69.4% of iterations). | Carbetocin is likely to result in better clinical outcomes and a modest cost-saving compared to oxytocin, although there is uncertainty. |
| Wohling and colleagues [40] | 1. Carbetocin (100 μg IV)<br>2. Oxytocin (5–10 IU, IV slow push) | • Average cost per patient was reduced by $63.46 AUD, year not stated ($57.88 USD 2023)<br>• Reduced rates of PPH >1,000 ml, 7.8% vs. 9.7% (OR 0.79, 95% CI 0.59–1.05).<br>• Significantly reduced rates of PPH >500 ml, 27.3% vs. 39.4% (OR 0.57, 95% CI 0.49–0.68).<br>• Reduced rates of additional treatment (26.9% vs. 46.9%).<br>• No difference in transfusions.<br><br>First year of data collection (2008) used for cost conversion calculation. | Carbetocin dominant compared to oxytocin. | Carbetocin conferred an absolute cost reduction, but more detailed cost analyses should be completed in the future. |
| Caceda and colleagues [42] | 1. Carbetocin as first line for prevention<br>2. Oxytocin as first line for prevention | Using carbetocin in place of oxytocin resulted in incremental health gains for incremental costs:Base Case<br>• S/49,918, PEN 2015, ($37,314.47 USD 2023) per QALY gained.<br><br>OWSA<br>• Results reported as robust.<br><br>PSA<br>• S/119,178 ($89,087.37 USD 2023) per QALY gained. | Carbetocin cost-effective compared to oxytocin at author's stated threshold of S/132,699 (3 × GDP per capita).<br>In USD 2023 this is $99,194.52. | Carbetocin more cost-effective than oxytocin for prevention of PPH after CS in this setting. |
| Henriquez-Trujillo and colleagues [43] | 1. Carbetocin<br>2. Oxytocin | Base case:<br>• ICER $2,432.89 USD 2015 ($2,831.87 USD 2023) per DALY averted.<br>PSA:<br>• Mean ICER $3,387.69 ($3,943.26 USD 2023) per DALY averted.<br>• 95% CI: $3,307.18–3,468.20 ($3,849.55–4,036.97 USD 2023).<br>• 97% of the ICER iterations were deemed very cost-effective. | Carbetocin cost-effective compared to oxytocin at author's stated threshold of $6,302 (1× GDP per capita).<br>In USD 2023 this is $7,335.50. | For the primary prevention of PPH after CS in Ecuador, carbetocin is highly cost-effective for both elective and emergency deliveries. |
| Voon and colleagues [45] | 1. Carbetocin (100 μg IV)<br>2. Oxytocin (5 IU IV) | Base case:<br>• Carbetocin would avert 108 episodes of PPH, 104 episodes of transfusion and the need for 455 patients to receive additional uterotonics (per cohort of 3,000).<br>• Carbetocin more expensive than oxytocin, ICER of $278.70 USD 2016 ($320.90 USD 2023) to avert 1 episode of PPH. | No threshold stated. | Authors described the ICER as favourable, but stated utilisation would depend on the value an institution places on averting re-treatment and their allocation of human resources. |

(*Continued*)

**Table 2.** (Continued)

| Study | Intervention/s and comparator/s (dose and route if specified) | Results | Dominance/cost-effectiveness | Summary of study conclusions |
|---|---|---|---|---|
| Higgins and colleagues [39] | 1. Carbetocin (100 μg IV) 2. Oxytocin (5 IU) | • No difference in frequency of PPH observed. • Increase in average costs per delivery of £18.52 GBP, year not stated ($33.59 USD 2023). Year of data collection (2010) used for cost conversion calculation. | Carbetocin neither dominant nor cost-effective compared to oxytocin. | Replacing oxytocin with carbetocin for PPH prophylaxis after CS is not beneficial to the patient or the delivery unit. |
| Matthijsse and colleagues [46] | 1. Carbetocin (100 μg IM) 2. Oxytocin (10 IU bolus) | Base case: • Carbetocin was less costly, -£55 GBP 2019 ($85.55 USD 2023), and more effective with 0.0342 less PPH events, or 0.0001 QALYs gained per woman. OWSA: • Results were robust to all variations tested. PSA: • Carbetocin dominated in 79.5% of runs. | Carbetocin dominant compared to oxytocin. | Carbetocin was found to cost-effective from a UK NHS perspective for the prevention of PPH following VB. |
| Vlassoff and colleagues [47] | 1. Oxytocin Uniject (10 IU IM) 2. Misoprostol (600 μg PO) 3. Standard care | • ICER of using misoprostol compared to standard care was $38.96, USD 2013 ($46.70 USD 2023), per PPH averted. • ICER of using oxytocin compared to standard care was $119.15 ($142.83 USD 2023) per PPH case averted. (Note: this model was based on a single RCT where 0% of women treated with misoprostol had PPH.) | Misoprostol dominant compared to oxytocin (more effective and lower cost). Misoprostol compared to standard care incurred incremental costs, and no threshold was stated. | In settings where a significant proportion of births take place outside of health facilities, without skilled providers, misoprostol-based PPH prevention could be cost-effective and improve maternal health. |
| Diaz and colleagues [48] | 1. Implementation of AMTSL with oxytocin, equipment upgrades, and staff training. 2. No active implementation of AMTSL, equipment upgrades, or staff training. | • Estimated 15,335 cases of PPH averted between 2001 and 2005. • Incremental cost of $3,328 USD, year not stated ($5,087 USD 2023) per averted case >500 ml or $29,897 ($45,703 USD 2023) per averted case >1,000 ml. First year of data collection (2001) used for cost conversion calculation. | No threshold stated. | The programme reduced the incidence of PPH, the main cause of maternal mortality. |
| Tsu and colleagues [49] | 1. Oxytocin Uniject (10 IU) 2. Oxytocin (10 IU drawn from ampoules) 3. No AMTSL | Base case • $15.70, USD 2004 ($22.59 USD 2023), per case of PPH averted with ampoules compared to no AMTSL. • $21.68 ($31.19 USD 2023), per case of PPH averted with Uniject compared to no AMTSL. Best case scenario • $7 ($10.07 USD 2023), per death averted with ampoules compared to no AMTSL. • $260 ($374.07 USD 2023), per death averted with Uniject compared to no AMTSL. Worst case scenario • $2,508 ($3,608.29 USD 2023), per death averted with ampoules compared to no AMTSL. • $3,463 ($4,982.26 USD 2023), per death averted with Uniject compared to no AMTSL. | No threshold stated. | The low net incremental cost of AMTSL suggests that the introduction of AMTSL in primary-level facilities in Vietnam can reduce the incidence of PPH and benefit women's health without adding much to national health care costs. |

*(Continued)*

**Table 2.** (Continued)

| Study | Intervention/s and comparator/s (dose and route if specified) | Results | Dominance/cost-effectiveness | Summary of study conclusions |
|---|---|---|---|---|
| Pichon-Riviere and colleagues [50] and correction [101] | 1. Oxytocin (10 IU IM or 5 IU IV drawn from ampoules) <br> 2. Oxytocin Uniject (10 IU IM) | • Uniject reduced PPH events and deaths, and increased QALYs in all 30 countries analysed. <br> • Incremental QALYs gained per 1,000 institutional deliveries ranged from 0.02 to 0.71. <br> • In 27% of the countries, Uniject was cost saving. In the remaining 22 countries, Uniject was associated with an incremental cost between $0.005 and $0.85, USD 2013 ($0.006 to $1.02 USD 2023), per delivery. <br><br> (Note: this model assumed that utilizing Uniject would increase the proportion of deliveries with access to Oxytocin.) | Uniject was cost-effective compared to ampoules in all 30 countries at the author's stated threshold (3 × GDP per capita of that country) with many likely cost-effective at lower thresholds (1× GDP per capita). | Uniject was modelled as either cost-saving or very cost-effective in almost all countries in Latin America and the Caribbean. Even if countries achieve only small increases in oxytocin use by incorporating Uniject, this strategy could be considered an efficient use of resources. |
| Carvalho and colleagues [51] | 1. Inhaled oxytocin (note: not yet a licensed product) <br> 2. Standard of care in that country and setting (different uterotonics depending on the setting) | Bangladesh Base case <br> • IHO introduction would avert over 18,500 PPH cases and 76 maternal deaths annually and save $716,000, USD 2017 ($809,031.98 USD 2023), per year. <br> OWSA <br> • Results were cost saving in all but one of the scenarios tested. <br> Ethiopia Base case: <br> • IHO introduction would avert 3,000 PPHs annually and 30 maternal deaths at an incremental cost of $1,443,000 ($1,630,493.22 USD 2023). <br> • $464 per PPH averted (524.29 USD 2023) <br> • $47,557 per maternal life saved ($53,736.22 USD 2023). <br> OWSA: <br> • IHO was not cost-effective in any of the OWSA scenarios. | Bangladesh: Inhaled oxytocin dominant compared to standard care. <br> Ethiopia: <br> Inhaled oxytocin not cost-effective at author's stated thresholds. | In settings like Bangladesh, where there is limited access to oxytocin, Inhaled Oxytocin could be a cost-saving intervention, as health impacts are accompanied by a substantial reduction in spending on PPH treatment. In the Ethiopian context, the product may not be considered a cost-effective intervention until the product is fully integrated into the health system. |
| Sutherland and colleagues [56] | 1. Misoprostol (600 µg administered by VHW). <br> 2. VHW attendance but no uterotonic. | • Intervention resulted in 38% (95% CI, 5%–73%) reduction in maternal deaths. <br> • ICER of $1,401 USD 2008 ($1,812.44 USD 2023) per life saved. <br> • IQR of ICER $1,008–$1,848 ($1,304.02–2,390.71 USD 2023). | No threshold stated. | Authors conclude that misoprostol is cost-effective and could potentially save tens of thousands of lives each year at low cost. |
| Sutherland and colleagues [57] <br> Note: this study appears in both prevention and treatment sections of this review. | 1. Misoprostol prevention: 600 µg oral misoprostol, if they haemorrhage >1,000 ml they have 75% chance of referral to health centre. <br> 2. Misoprostol treatment: 800 µg sublingual misoprostol after 700 ml blood loss. <br> 3. Standard care with unskilled assistant and no medication. | • Misoprostol for PPH prevention was the most effective and costly intervention. <br> • In addition to the DALYs averted and costs incurred in the misoprostol for PPH treatment strategy, misoprostol prevention would avert a further 33.6 DALYs and incur an additional cost of $5,721 USD 2009 ($7,345.09 USD 2023) per 10,000 deliveries. <br> • ICER of $170 ($218.26 USD 2023) per DALY averted. | Misoprostol prevention is very cost-effective compared to alternatives at author's stated threshold of $2,600 (1× GDP per capita). This is $3,338.10 in USD 2023. | Misoprostol for prevention is very cost-effective for decreasing mortality and anaemia compared to standard care. |
| Goldie and colleagues [52] | 1. Misoprostol distribution in community (in home and birthing centres) in addition to general infrastructure and service upgrades. <br> 2. Implementing service and infrastructure upgrades without adding misoprostol distribution. | Intervention resulted in <br> • Cost savings of $120–$198 million USD 2006 ($162 to $268 million USD 2023) over a lifetime time horizon. <br> • Reduction in maternal deaths of 6.9%–12.3%. | Misoprostol strategy dominated alternative. | Although not a substitute for reliable obstetric care, community-based distribution of oral misoprostol in homes and birthing centres is likely to be cost-effective intervention. |

(Continued)

**Table 2.** (Continued)

| Study | Intervention/s and comparator/s (dose and route if specified) | Results | Dominance/cost-effectiveness | Summary of study conclusions |
|---|---|---|---|---|
| Lubinga and colleagues [54] and correction [102] | 1. Misoprostol (600 μg PO distributed to women in their antenatal visit or as part of a safe delivery kit). 2. Oxytocin (10 IU IM but limited to only those they deliver in a facility). | Base case: <br>• The intervention had an ICER of $181 USD 2012 ($220.77 USD 2023) per DALY averted from a government perspective, and $64 ($78.06 USD 2023) per DALY averted from a modified societal perspective. <br><br>PSA: <br>• The ICER ranged from $81 to $441 ($98.80 to $537.91 USD 2023) per DALY averted from the government and $-84 to $260 ($-102.46 to 317.13 USD 2023) per DALY averted from the societal perspective. <br>• 100% of the iterations were below the author's cost-effectiveness threshold from both perspectives. | Misoprostol strategy cost-effective compared to oxytocin strategy at author's stated threshold of $1,641 (3× GDP per capita). This is $2,001.59 in USD 2023. | Prenatal distribution of misoprostol could potentially save lives at modest incremental costs in this setting. |
| Prata and colleagues [55] Note: Only the comparison of ANC vs. ANC-miso interventions met inclusion criteria for this review. | 1. Antenatal and postpartum care as outlined in the "WHO Mother Baby Package". 2. Implementing the same as above, plus community distribution of misoprostol for home births. | Low infrastructure settings: <br>• Adding misoprostol to MBP antenatal care would cost an extra $4,900.42 USD 2007 ($6,462.86 USD 2023) and avert an extra 23 maternal deaths per cohort of 500,000 (calculated from Table 1). <br><br>Medium infrastructure settings: <br>• Adding misoprostol to MBP antenatal care would cost an extra $5,568.64 ($7,344.13 USD 2023) and avert an extra 23 maternal deaths per cohort of 500,000 (calculated from Table 2). <br><br>High infrastructure settings: <br>• Adding misoprostol to MBP antenatal care would cost an extra $5,647.34 ($7,447.92 USD 2023) and avert an extra 15 maternal deaths per cohort of 500,000 (calculated from Table 3). | No threshold stated. | Authors conclude that family planning plus safe abortion services and antenatal care which includes the distribution of misoprostol for PPH prevention at home births are the two most cost-effective interventions. |
| Lang and colleagues [53] | Scenario 1: 1. Oxytocin (in hospitals) and misoprostol (in community). 2. Oxytocin (in hospitals and no treatment in the community). Scenario 2: 1. Misoprostol in both hospital and community. 2. No uterotonics in either hospitals or community settings. | Scenario 1: <br>• The intervention would avert 22 cases of PPH, 2 cases of severe PPH, the requirement for 6 women to have additional uterotonics, and 4 women to have transfusions. <br>• The intervention would result in an additional 130 women would experience shivering and 42 women fever. <br>• This intervention would save $320, USD 2012, ($390.32 USD 2023) for the cohort of 1,000 women. <br><br>Scenario 2: <br>• The intervention would avert 37 cases of PPH, 3 cases of severe PPH, the requirement for 10 women to have additional uterotonics, and 6 women to have transfusions. <br>• The intervention would result in an additional 217 women experiencing shivering and an extra 70 women fever. <br>• The intervention would save $533 ($650.12 USD 2023) per cohort of 1,000 women. | Scenario 1: Oxytocin in hospitals and misoprostol in community dominates oxytocin in hospitals alone. Scenario 2: Using misoprostol in both community and hospital dominates no uterotonic use. | Even though misoprostol is not the optimum choice in the prevention of PPH, it could be an effective and cost-saving choice where oxytocin is not or cannot be used for a variety of reasons. |

*(Continued)*

**Table 2.** (Continued)

| Study | Intervention/s and comparator/s (dose and route if specified) | Results | Dominance/cost-effectiveness | Summary of study conclusions |
|---|---|---|---|---|
| Fullerton and colleagues [58] | 1. AMTSL (uterotonic not specified)<br>2. EMTSL | Guatemala:<br>• 100 maternal deaths averted and net cost-saving of $ 18,000 USD 2004 ($25,896.82 USD 2023) per 100,000 births.<br>• Results were robust to a wide range of OWSA scenarios.<br><br>Zambia:<br>• 67 maternal deaths averted and net cost-saving of over $145,000 ($208,613.23 USD 2023) per 100,000 births.<br>• Results were robust to a wide range of OWSA scenarios. | AMTSL strategy dominated EMTSL strategy in both settings. | AMTSL is associated with a distinct financial benefit to health facilities in addition to clinical benefits. |
| Dazelle and colleagues [59] | 1. Prophylactic TXA (1 g) to all women<br>2. Prophylactic TXA (1 g) to women at high risk of PPH<br>3. Prophylactic TXA (1 g) to women at high or moderate risk of PPH<br>4. Routine care | Base case:<br>• All TXA strategies were considered superior in cost-savings and outcomes averted relative to routine care.<br>• The most effective strategy was providing TXA to all births which resulted in cost-savings of $690 million USD 2020 ($734 million USD 2023) and prevented 149,505 PPH cases, 19,447 balloon tamponades, 24,079 uterus-sparing surgeries, 2,933 hysterectomies, and 70 maternal deaths per year.<br><br>OWSA:<br>• The results were robust to variation in the value of all assessed variables.<br><br>PSA:<br>• Prophylactic TXA strategies are cost-saving versus the status quo in >99.9% of simulations. | All interventions dominated routine care. | Routine TXA prophylaxis for delivering women in the US is likely to result in substantial cost-savings and reductions in maternal morbidity and mortality. |
| Durand-Zaleski and colleagues [60] | 1. TXA (1 gram slowly IV over 2 min) in addition to standard care.<br>2. Placebo and standard care. | • No statistically significant difference in costs between the intervention and comparator.<br>• No statistically significant difference in rates of PPH>500 ml.<br>• The number of "provider-assessed clinically significant postpartum haemorrhage" was reduced (7.8% vs. 10.4%) and the proportion requiring secondary uterotonics was also lower (7.2% vs. 9.7%).<br><br>PSA:<br>• 65%–73% probability that TXA is both cost-saving and event-reducing. | TXA dominated routine care. | Prophylactic use of TXA at VB reduces both costs and bleeding events with a probability greater than 60% but less than 80%. |
| Sentilhes and colleagues [61] | 1. TXA in addition to standard care.<br>2. Placebo in addition to standard care. | • Mean length of stay was slightly lower for intervention group (4.8 days vs. 4.9 days).<br>• Proportion of women with no complication up to 90 days was higher in intervention group (70.7% vs. 66.0%).<br>• Mean total cost (until the occurrence of complications) were slightly higher in the intervention group, €3,321 vs. €3,260, EUR 2019 ($4,820.95 vs. $4,732.40, USD 2023).<br>• ICER of €762 ($1,106.16 USD 2023) per additional CS delivery without complication at day 90. | TXA strategy had 99.9% probability of being cost-effective at the author's stated threshold of €10,000 per additional CS delivery without complication at day 90. | Prophylactic use of tranexamic acid is cost-effective for reducing complications, including PPH, in women undergoing CS. However, the overall difference in effectiveness and costs between the two groups is low. |

(*Continued*)

**Table 2.** (Continued)

| Study | Intervention/s and comparator/s (dose and route if specified) | Results | Dominance/cost-effectiveness | Summary of study conclusions |
|---|---|---|---|---|
| Denison and colleagues [64,65] | 1. Two puffs of GTN (400 µg sublingual)<br>2. Placebo spray (sublingual) | • GTN was not effective at reducing the requirement for manual removal of placenta or the risk of PPH. | Not dominant nor cost-effective. | The GTN group incurred higher costs (not statistically significant) for no improvement in clinical, safety or patient-oriented outcomes. |
| Sharma and colleagues [66] | 1. Negative intrauterine pressure suction device integrated with AMTSL (with oxytocin).<br>2. AMTSL (with oxytocin) alone. | • Women that received the intervention had on average less blood loss (216.66 ml vs. 389.45 ml).<br>• A lower proportion of women receiving the intervention had PPH (0.49% vs. 1.81%, $p < 0.001$).<br>• Women treated with AMTSL alone incurred higher costs secondary to blood product use, no other costs were considered. | No threshold stated. | In low resource settings, adding the use of negative intrauterine pressure suction devices can be instrumental in decreasing the incidence of PPH. However, further large multi-centre trials are required before drawing solid conclusions. |
| Hong and colleagues [67] | 1. Internal iliac artery balloon occlusion prior to CS. Balloon inflated once baby delivered. Deflated after haemostasis achieved.<br>2. No occlusion before CS. | • The intervention was not effective at reducing the rates of any of the health outcomes of interest including blood loss or rate of hysterectomy but did increase the duration of surgery and the cost of hospitalization: ¥45,117 ± ¥9,359 vs. ¥30,615 ± ¥11,587, CNY, year not stated ($14,839 ± $3,078 vs. $10,069 ± $3,811 USD 2023).<br><br>First year of data collection (2015) used for cost conversion calculation. | Not cost-effective (no threshold set by author, but intervention not clinically effective). | Conclusions were not able to be drawn due to the heterogeneity of cases involved, retrospective study design, and results that showed little difference between intervention and comparator. |
| Niola and colleagues [68] | 1. Intravascular uterine artery occlusion immediately before delivery.<br>2. Surgery or embolization after delivery. | • Intervention group had lower rates of transfusion (36% vs. 100%) and lower rates of hysterectomy (26% vs. 43.4%).<br>• Intervention group had lower average costs per patient, €7,607.77 vs. €13,925.04, EUR, year not stated ($12,526.40 vs. $22,927.95 USD 2023).<br><br>First year of data collection (2009) used for cost conversion calculation. | Intervention dominant compared to standard postdelivery embolization. | In this sample of women with placental implant anomalies, predelivery uterine artery embolization was a cost-effective procedure. |
| Xue and colleagues [69] | 1. MDT-ERAS intervention (a multimodal perioperative care pathway to achieve enhanced recovery after surgery).<br>2. Traditional perioperative care. | • The intervention group experienced statistically significant lower proportion of PPH (5.94% vs. 17.13%, $P < 0.01$).<br>• Average cost of hospitalization was also reduced in the intervention group, ¥3,552.39 vs. ¥3,880.47 CNY, year not stated ($1,118.22 vs. $1,221.49 USD 2023).<br><br>First year of data collection (2018) used for cost conversion calculation. | MDT-ERAS dominated standard care (however, hospitalization was the only cost considered). | MDT-ERAS for CS may improve clinical outcomes and the costs of admission. |

Economic results are stated as they appear in the original publications and as a conversion to USD 2023 using an online tool developed by the Campbell and Cochrane Economics Methods Group (CCEMG) and the Evidence for Policy and Practice Information and Coordinating Centre (EPPI-Centre) [19]. Cost conversions were completed in December 2023 and may change slightly depending on final GDP figures.

AMTSL, active management of the third stage of labour; ANC, antenatal care; AUD, Australian dollars; CAD, Canadian dollars; CI, confidence interval; COP, Colombian peso; CNY, Chinese Yuan; CS, cesarean section; DALY, disability adjusted life year; EMTSL, expectant management of the third stage of labour; EUR, Euro; GBP, British Pound; GDP, gross domestic product; GTN, glyceryl trinitrate; HTA, health technology assessment; ICER, incremental cost-effectiveness ratio; IHO, inhaled oxytocin; IM, intramuscular; IU, international unit; IQR, interquartile range; IV, intravenous; MDT-ERAS, multidisciplinary team enhanced recovery after surgery; MBP, mother baby package; NHS, National Health Service; OR, odds ratio; OWSA, one-way sensitivity analysis; PEN, Peruvian sol; PO, per oral; PPH, postpartum haemorrhage; PRBC, packed red blood cells; PSA, probabilistic sensitivity analysis; QALY, quality-adjusted life year; RCT, randomised control trial; SE,: side effects; TXA, tranexamic acid; USD, United States Dollar; VHW, Village Health Worker; VB, vaginal birth; µg, microgram.

completeness of methods, prevented meaningful comparisons between studies. One of the studies was on an inhalable form of oxytocin still under investigation, and the other 4 studies were each based on effectiveness estimates inconsistent with more recent evidence. For completeness, the results of each of these studies are summarised in S11 Appendix; however, the most complete assessment of the cost-effectiveness of oxytocin appear as part of the UK health technology assessment from 2019 discussed earlier [30,31].

Six modelling studies analysed cost-effectiveness of misoprostol [52–57]. An Indian study assessed preventative misoprostol administered by village health workers (VHW) during home births compared to no uterotonics. They reported this strategy could reduce maternal deaths by 38% and cost $1,812.44 per death averted [56]. Another comparing misoprostol as prevention or treatment to no uterotonics found preventative misoprostol was the most effective strategy, incurring $218.26 per disability-adjusted life year (DALY) averted compared with misoprostol as treatment [57]. A third Indian study considered adding community-based distribution of misoprostol to other service and infrastructure upgrades [52]. The estimated maternal mortality reduction compared to other upgrades alone was modest (6.9% to 12.3%), but estimated to be cost-saving over a lifetime horizon [52]. These 3 studies assumed misoprostol to be significantly more effective than indicated in systematic reviews (Table O in S8 Appendix) and may therefore overestimate its cost-effectiveness [30,37].

In other settings, a Ugandan study estimated advanced misoprostol distribution would improve health outcomes at small incremental costs from the governmental perspective ($98.80 to $537.91 per DALY averted) or dominate under a societal perspective [54]. An analysis across sub-Saharan Africa similarly concluded community misoprostol use would incur additional costs and reduce maternal deaths [55]; they described the strategy as cost-effective without specifying a threshold. The final study assessed community or hospital misoprostol usage in international settings where injectable uterotonics are unavailable [53]. They concluded that either strategy of utilising misoprostol in the community alone, or in both hospitals and community, would be dominant [53]. As with the previous misoprostol studies, the underlying effectiveness estimates of all 3 studies was inconsistent with systematic reviews (Table O in S8 Appendix) [30,37].

A single modelling study analysed active management of the third stage of labour in place of expectant management in Guatemala and Zambia, with an unspecified uterotonic [58]. In both settings the authors estimated that active management of the third stage of labour would reduce maternal deaths and decrease health facility costs [58].

**Tranexamic acid for prevention.** Three high-quality studies assessed the impact of tranexamic acid for PPH prevention in high-income settings, 1 modelling study in women (vaginal birth or cesarean section) in the United States [59], and 2 economic evaluations of effectiveness studies in either women undergoing vaginal birth [60] or cesarean section [61] in France. In all 3 studies, the tranexamic acid strategy was either deemed cost-effective [61] or dominant [59,60]; however, there were some differences in the results. In the United States-based model, the impact on cost savings and avoided morbidity were substantial, while in the French studies, there were little differences in the overall effectiveness and costs between tranexamic acid and standard care strategies. The 3 studies scored high on CHEC-E, and 2 were based on effectiveness estimates either consistent [59] with systematic review estimates [7,62,63], or may have underestimated the effectiveness of tranexamic acid [60], thereby providing conservative cost-effectiveness estimates. The effectiveness estimates in one study were unable to be assessed, as they used a composite effectiveness measure not reported in previous systematic reviews [61].

**Glyceryl trinitrate for prevention.** A health technology assessment completed in UK hospitals evaluated the use of glyceryl trinitrate for the management of retained placenta to reduce

the requirement for invasive procedures and the risk of PPH [64,65]. It was not effective at reducing the rate of manual placenta removal, or subsequent PPH, and was therefore not cost-effective [64,65].

**Devices for prevention.** One study assessed the cost-effectiveness of using a negative intrauterine pressure suction device alongside active management of the third stage of labour to minimise blood loss and prevent PPH in women following low risk vaginal births in India [66]. The study reported that using the device resulted in reduced blood loss, reduced rates of PPH (0.49% versus 1.81%, $p < 0.001$) and reduced hospital expenditure on blood products. Although this study met inclusion criteria for this review, the health economic analysis was extremely limited and only costed blood product usage. In addition, the underlying effectiveness evidence for negative intrauterine pressure devices remains uncertain and large-scale multicentre clinical trials are required [66].

**Surgical practices for prevention.** The remaining 3 evaluations of preventative interventions related to surgical methods for minimising PPH [67–69]. The effectiveness of these interventions is debated. In addition, all 3 scored low on CHEC-E and pertained to a small proportion of women at risk of PPH (those with placenta accreta spectrum and those at high risk of PPH undergoing cesarean section). We therefore summarised their results separately in S12 Appendix.

### Economic evaluations of diagnostic interventions

One study in a USA hospital examined the effects of introducing the Triton system—combining colorimetric and gravimetric analysis for quantitative assessment of postpartum blood loss at all births—to improve PPH diagnosis (Table 3) [70]. Triton implementation increased the

**Table 3. Results of diagnostic intervention studies.**

| Study | Intervention/s and c comparator/s (dose and route if specified) | Results | Dominance/cost-effectiveness | Summary of study conclusions |
|---|---|---|---|---|
| Katz and colleagues [70] | 1. Triton system for blood measurement (gravimetric and colorimetric). 2. Visual estimation by obstetrician, nursing staff, or anaesthetist. | • The intervention group reported a higher proportion of deliveries diagnosed with PPH in both VB (2.2% vs. 0.5%) and CS (12.6% vs. 6.4%). <br>• The intervention group recorded less blood loss on average in VB (300 ml vs. 258 ml) and CS (800 ml vs. 702 ml). <br>• The intervention group reported higher secondary use of uterotonics (22% vs. 17.3%). <br>• No difference in the use of blood products between groups. <br>• No difference between groups for blood bank costs. <br>• On average laboratory costs were $4 USD, year not stated, cheaper per patient in the intervention ($4.61 USD 2023). <br><br>First year of data collection (2016) used for cost conversion calculation. | Not calculated. | Use of quantitative blood measurement in this setting resulted in increased vigilance at VB and CS and improved the identification of PPH and the secondary use of uterotonics. The authors also state that the cost saving from reduced laboratory costs would result in 152% return on investment, but this analysis does not include many other costs. |

Economic results are stated as they appear in the original publications and as a conversion to USD 2023 using an online tool developed by the Campbell and Cochrane Economics Methods Group (CCEMG) and the Evidence for Policy and Practice Information and Coordinating Centre (EPPI-Centre) [19]. Cost conversions were completed in December 2023 and may change slightly depending on final GDP figures.

CS, cesarean section; PPH, postpartum haemorrhage; USD, United States Dollar; VB, vaginal birth.

proportion of women diagnosed with PPH. They calculated it would result in savings, though the analysis focussed on few types of cost.

### Economic evaluations of treatment interventions

**Medical interventions for PPH treatment.** Two studies assessed PPH treatment with misoprostol in settings where other treatment options are limited (Table 4) [57,71]. The first analysed the cost-effectiveness of training traditional birth attendants (TBAs) to recognise and treat PPH with per-rectal misoprostol in sub-Saharan Africa [71]. They found it could prevent 1,647 cases from progressing to severe PPH, simultaneously saving $160,922 per 10,000 births [71]. The other study in India assessed a similar strategy of unskilled birth attendants diagnosing and treating PPH with misoprostol [57]. In this case, the health benefits (not as large as preventative strategies but preferable to no intervention) incurred a low incremental cost of $7.70 per DALY averted [57]. The studies were moderate [57] to high [71] quality. Bias related to treatment effects could not be assessed (Table U in S8 Appendix).

Four modelling evaluations assessed tranexamic acid for PPH treatment—2 in the USA [72,73] and 2 in LMICs (India [74], Nigeria, and Pakistan [75]). Both USA studies considered women with PPH following either vaginal birth or cesarean section, concluding that tranexamic acid would likely avert laparotomies, deaths, and reduce healthcare and societal costs, whether on a short-term time horizon [73] or a lifetime horizon [72]. The LMIC-based studies were similarly positive for health outcomes but estimated modest incremental costs. These ranged from $86.03 per QALY gained in India [74] to $239.49 in Nigeria [75]. All 4 evaluations were high quality, and the effectiveness estimates used were consistent with systematic reviews.

**Devices for PPH management.** One evaluation of an effectiveness study [76] and 1 modelling study [77] assessed non-pneumatic anti-shock garment (NASG) for women with obstetric haemorrhage in LMICs. One study in Zimbabwe and Zambia compared the cost-effectiveness of applying NASGs to patients in primary health centres awaiting definitive management at a referral centre, to waiting until patients arrived at the referral centre to apply NASG [76]. Results differed between sites, but on average the costs of distributing and training users of NASG were largely offset by the reduction in other health resources utilised when delaying application. A small incremental cost of $27.64 was incurred for each DALY averted by the intervention [76]. The second study in Egypt and Nigeria assessed the use of NASG within tertiary hospitals, against routine care without the temporising device [77]. In Egypt, NASG improved health outcomes and reduced costs, while in Nigeria it improved health outcomes at a modest incremental cost of $3.97 per DALY averted [77]. These evaluations were moderate [76] to high [77] quality. Bias related to treatment effects could not be assessed (Table U in S8 Appendix).

Two modelling studies assessed the cost-effectiveness of treating PPH with UBT devices in Kenya and India [78,79]. The study in Kenya assessed the "Every Second Matters" (ESM) UBT against standard care either with or without any form of uterine packing [79]. In this setting, utilising ESM-UBT was considered cost-effective compared to standard care, regardless of whether uterine packing was able to be performed. Incremental costs were between $30.26 and $231.64 per DALY averted, depending on the unit cost and which comparator was chosen [79]. This model was moderate quality and may be at risk of bias related to treatment effects (Table U in S8 Appendix). The Indian study compared the relative cost-effectiveness between 3 UBT models—the ESM-UBT, the Bakri-UBT, or an improvised condom-UBT [78]. Although the base case of this analysis estimated the usage of the ESM-UBT to be the dominant strategy, the probabilistic sensitivity analysis (PSA) showed large uncertainty, and the

**Table 4. Results of treatment intervention studies.**

| Study | Intervention/s and comparator/s (dose and route if specified) | Results | Dominance/cost-effectiveness | Summary of study conclusions |
|---|---|---|---|---|
| Bradley and colleagues [71] | 1. Training of TBAs to recognise PPH and treat with misoprostol (1,000 µg PR). 2. TBA attends birth but refers patient to hospital if PPH occurs. | • Training TBAs and giving misoprostol if needed would prevent 1,647 cases of severe PPH (810–2,920 in sensitivity analysis). • This would save $115,336 USD 2005 ($160,922.36 USD 2023) on transport, hospital fees, IV therapy, and blood products. • The uncertainty range for the estimated savings was $13,991 to $1,563,593 ($19,521–$2,181,600 USD 2023). | Training TBAs and utilizing misoprostol dominated alternative. | This intervention has the potential to save millions of dollars and improve maternal health in settings with limited health resources. |
| Sutherland and colleagues [57] | 1. Misoprostol prevention: 600 µg PO. If patient has haemorrhage >1,000 ml they have 75% chance of referral to health centre. 2. Misoprostol treatment: 800 µg sublingual after 700 ml blood loss. 3. Standard care with unskilled assistant and no medication. | • Misoprostol for PPH treatment would save an additional 216 DALYs and would incur an additional cost of $1,212 USD 2009 ($1,556 USD 2023), per 10,000 deliveries. • ICER of $6 ($7.70 USD 2023) per DALY averted when compared to standard management with unskilled assistant alone. • Not as effective as using misoprostol for prevention of PPH which could potential save an extra 33.6 DALYs per 10,000 deliveries. | Misoprostol treatment very cost-effective compared to standard care at author's stated threshold of $2,600 (1× GDP per capita). This is $3,338 in USD 2023. | Misoprostol for treatment of PPH is very cost-effective for decreasing mortality and anaemia compared to standard care. |
| Howard and colleagues [72] | 1. TXA (1 g) plus standard management 2. Standard care | Base case: • Intervention would avert 6 exploratory laparotomies following VB, 112 reoperations after CS, and 11 deaths per 100,000 deliveries. • This is a gain of 329 QALYs. • Intervention would save $15.39 million, USD 2019, annually. ($16.71 million USD 2023) • If administered early (<3 h) intervention would gain 438 QALYs and save $23.15 million ($25.13 million USD 2023) annually per 100,000 deliveries. PSA: • Early administration of TXA for the treatment of PPH was dominant in 99.8% of samples. | TXA dominated standard care. | TXA is a cost-effective strategy for reducing morbidity and mortality from PPH in the USA. |
| Sudhof and colleagues [73] | 1. TXA given at any time. 2. TXA given within 3 h of delivery. 3. Standard care (no TXA). | Base case: • Intervention saved $11.3 million, USD 2018 ($12.49 million USD 2023), prevented 334 laparotomies, and averted 9 maternal deaths in the USA annually assuming 4 million births with a 3% rate of PPH. • Giving TXA <3 h from delivery almost tripled the cost savings and improved maternal outcomes much further. For an annual US cohort, it would prevent 924 more laparotomies, 5 additional maternal deaths, and saved an additional $18.8 million ($20.77 million USD 2023). PSA: • TXA strategies were cost saving in >99.9% of simulations. | Both TXA strategies dominated the alternative with TXA <3 h being the most dominant. | Routine TXA early in the treatment of PPH is likely to be cost saving in the United States. |

*(Continued)*

**Table 4.** (Continued)

| Study | Intervention/s and comparator/s (dose and route if specified) | Results | Dominance/cost-effectiveness | Summary of study conclusions |
|---|---|---|---|---|
| Joshi and colleagues [74] | 1. TXA (1 g IV within 3 h of birth) plus standard care. An additional dose of TXA was given if bleeding continued after 30 min or if it restarted within 24 h.<br>2. Standard care | **Base case**<br>• Intervention would incur ₹121 INR 2019/20 ($7.08 USD 2023) and gain 0.082 QALYs per woman treated.<br>• This equates to an ICER of ₹1,470 ($86.03 USD 2023) per QALY gained.<br>• Per annual cohort of 510,915 with PPH, intervention would avert 905 surgeries, 655 ICU admissions, and 1,990 maternal deaths compared to standard care.<br><br>**PSA:**<br>• For an Indian willingness to pay threshold value of one-time GDP per capita, the analysis suggested that 94.5% of simulations are cost-effective. | TXA cost-effective compared to standard care at author's stated threshold of 1× GDP per capita ₹145,742 ($8,529 USD 2023). | Early administration of TXA to women with PPH in Indian public health facilities is recommended from a cost-effectiveness perspective. |
| Li and colleagues [75] | 1. TXA plus routine care<br>2. Placebo plus routine care | **Base case Nigeria**<br>• 0.18 QALYs gained for an additional cost of $37.12 USD 2016 ($42.74 USD 2023) per patient.<br>• This is an ICER of $208 ($239.49 USD 2023) per QALY gained.<br><br>**Base case Pakistan**<br>• 0.08 QALYs gained for an additional cost of $6.55 ($7.54 USD 2023) per patient.<br>• This is an ICER of $83 ($95.57 USD 2023) per QALY.<br><br>**PSA:**<br>• At the lower end of the threshold range for Nigeria, the probability that TXA is cost-effective is 93%.<br>• At the lower end of the cost-effective threshold range for Pakistan, the probability that TXA is cost-effective is 98%. | TXA cost-effective compared to standard care in both settings at author's stated threshold of $446–$2,880 per QALY in Nigeria and $314–$2,416 per QALY in Pakistan.<br>In USD 2023 this is $514–$3,316 per QALY in Nigeria and $362–$2,782 per QALY in Pakistan. | Early treatment of PPH with TXA is highly cost-effective in Nigeria and Pakistan and is likely to be cost-effective in countries in sub-Saharan Africa and southern Asia with a similar baseline risk of death due to maternal haemorrhage. |
| Downing and colleagues [76] | 1. Application of the NASG at the primary health care centre.<br>2. Delaying application of NASG until the patient arrives at the referral hospital. | • Early application group had 0.712 fewer DALYs than those in the later application group for an incremental cost of $15.51 international dollars 2010 ($19.68 USD 2023).<br>• Early NASG application costs $21.78 ($27.64 USD 2023) per DALY averted compared to delayed application.<br><br>**Sensitivity analysis**<br>• The ICER of applying the NASG early compared to later was sensitive to the unit cost of blood transfusions and ranged from $9.22–$87.85 ($11.70–$111.49 USD 2023). | Early application cost-effective compared to late application in Zambia.<br>Early application not cost-effective in Zimbabwe. | The evidence from Zambia supports the early application of NASG for women with hypovolemic shock from obstetric haemorrhage in the community setting. The evidence from Zimbabwe was suggestive of this too, but not statistically significant. |

*(Continued)*

**Table 4.** (Continued)

| Study | Intervention/s and comparator/s (dose and route if specified) | Results | Dominance/cost-effectiveness | Summary of study conclusions |
|---|---|---|---|---|
| Sutherland and colleagues [77] | 1. Adding NASG to standard management of women with severe hypovolemic shock (MAP < 60 mmHg) due to obstetric haemorrhage. 2. Adding NASG to standard management of women with any degree of shock due to obstetric haemorrhage. 3. Standard care with no NASG. | Egypt • Using the NASG for cases of severe shock resulted in decreased deaths, hysterectomies, and severe morbidity (357 DALYs averted) and saved $9,489 international dollars 2010 ($12,042 USD 2023), per 1,000 women with shock. • When applying NASG to all women with shock a further 37 DALYs were averted and an extra $21,253 ($26,972 USD 2023) were saved.<br><br>Nigeria • Using the NASG for cases of severe shock resulted in decreased deaths, hysterectomies, and severe morbidity (2,063 DALYs averted) for an incremental cost of $6,460 ($8,198 USD 2023). • This is an ICER of $3.13 ($3.97 USD 2023) per DALY averted. • Using NASG on all cases of shock was less effective and more costly than this. | Egypt Both applying NASG to women with any shock or severe shock dominated standard care. Nigeria Both applying NASG to women with any shock or severe shock was cost-effective compared to standard care, but applying to severe cases was most effective. | The NASG is either cost saving or highly cost-effective for women in severe hypovolemic shock when administered in a tertiary care setting. |
| Mvundura and colleagues [79] | 1. ESM-UBT plus standard care 2. Standard care without UBT or uterine packing 3. Standard care with uterine packing available | • ESM-UBT could prevent 1,255 hospital transfers, 430 hysterectomies, and 44 maternal deaths compared to no standard care, no packing. • The ICER for the UBT device was $26–$40 USD 2015 ($30.26–$46.56 USD 2023) per DALY averted based on UBT device cost of $5–$15. • If uterine packing is available, the ICER per DALY averted was $164–$199 ($190.90–$231.64 USD 2023) depending on UBT cost.<br><br>OWSA: • Results in both scenarios were robust to all parameter changes in the OWSA and remained cost-effective. | ESM-UBT was cost-effective compared to both alternatives at the authors stated threshold of $1,358 (1× GDP per capita in 2014). This is $1,581 in USD 2023. | ESM-UBT is a cost-effective way to reduce hospital transfers, surgeries, and maternal deaths caused by severe PPH. The results of this study could be used to guide the expansion of this intervention in Kenya and other similar settings. |
| Joshi and colleagues [78] | 1. ESM-UBT 2. Bakri-UBT 3. Condom-UBT (improvised) which is standard care | Base case: • ESM-UBT versus condom-UBT has an ICER value of ₹-2,412 INR 2017 (-$153 USD 2023) per DALY averted meaning an incremental cost-saving of ₹2,412 (or $153 USD 2023) occurs per incremental DALY averted. • Bakri-UBT was less effective and more costly than condom-UBT.<br><br>PSA: • Comparing ESM-UBT versus Condom-UBT: 63.5% of the simulations at the given willingness to pay threshold were cost-effective (52% dominant). | ESM-UBT dominated standard care with condom-UBT in 52% of simulations and was cost-effective in 63.5% at authors stated threshold of ₹24,211 (1,536 USD 2023). This was reported as a high degree of uncertainty. | Condom-UBT device as recommended for atonic PPH management in India offers better value as compared to Bakri-UBT in this setting. ESM-UBT could be a cost-saving alternative, but this needs further evaluation as the differences in costs and health outcomes are marginal, there is a high degree of uncertainty. |

(*Continued*)

**Table 4.** (Continued)

| Study | Intervention/s and comparator/s (dose and route if specified) | Results | Dominance/cost-effectiveness | Summary of study conclusions |
|---|---|---|---|---|
| Edwards and colleagues [80] | 1. Butterfly device (facilitates compression of uterus as an alternative to bimanual compression)<br>2. Standard care | Base case:<br>• Treatment with the intervention resulted in reduced PPH progression beyond 1,000 ml after device use (1.75% vs. 7.97%) compared to standard care.<br>• Mean ICER of £3,795.78 GBP 2017 ($6,126.91 USD 2023) per PPH progression avoided (defined as >1,000 ml blood loss after point of device use).<br><br>PSA:<br>• At a threshold of £8,500 ($13,720 USD 2023) per PPH progression avoided, the intervention has a probability of being cost-effective of 87%. | No comparable threshold available. | The Butterfly device is a relative low-cost device in a UK NHS setting with a high probability of being cost-effective. |
| Snegovskikh and colleagues [82] | 1. Blood product resuscitation guided by point of care viscoelastic testing (PCVT).<br>2. Empiric blood product resuscitation. | • Estimated blood loss, post-operative ICU admission, and the incidence of hysterectomy were significantly lower in the PCVT group.<br>• The average cost of hospitalization was lower for the patients in the PCVT group: $11,802.94 vs. $20,419.08 USD, year not stated ($14,672.66 vs. $25,383.69 USD 2023).<br><br>First year of data collection (2011) used for cost conversion calculation. | Only aggregate hospital costs presented—unclear if overall cost-effective assessment can be made. | PCVT-based protocols conferred a reduction in the need for PRBC, FFP, and platelet concentrate transfusions in the setting of severe PPH. This individualised approach may also result in less intraoperative blood loss, lower rates of puerperal hysterectomy and postoperative ICU admissions, as well as a reduce the length of hospital stay and cost of hospitalization. |
| Einerson and colleagues [81] | 1. Universal type and screen plus cross match for high-risk patients<br>2. Universal type and screen only<br>3. Universal hold clot plus cross match for high-risk patients<br>4. Selective type and screen only in high-risk patients<br>5. No routine admission testing | • Strategy 1 generated an ICER of $115,541 USD 2015 ($134,489 USD 2023) per emergency-release transfusion prevented compared with strategy 3 (the next most effective strategy).<br>• The ICER for strategy 3 was $2,878 ($3350 USD 2023) per emergency-release transfusion prevented compared with strategy 5. | Strategy 5 was the most cost-effective at the author's prespecified threshold of $1,500 ($1,746 USD 2023) to prevent one emergency-release transfusion (76.1% of PSA iterations). | Universal type and screen strategies were not cost-effective in a general obstetric population even when considering a wide range of assumptions, variable ranges, and willingness-to-pay thresholds. The small incremental gains in prevention of emergency-release transfusion were not offset by the added costs of the transfusion preparedness strategies. |
| Prick and colleagues [85] | 1. RBC transfusion aiming for target Hb 8.9 g/dl.<br>2. Conservative management (iron and or folic acid supplementation and only utilizing transfusion if clinically indicated). | • Transfusing to target Hb incurred an incremental cost of €431 EUR 2013 ($627 USD 2023) for each 1-point improvement on the Multidimensional Fatigue Inventory. | No threshold set. | In women with acute anaemia after PPH, RBC transfusion is on average €249 ($362 USD 2023) more expensive per woman than non-intervention, with only a small gain in fatigue scores following transfusion. A policy of non-intervention appears justified for women in these circumstances (Hb in range 4.8–7.9 g/dl and no symptoms of severe anaemia). |

*(Continued)*

**Table 4.** (Continued)

| Study | Intervention/s and comparator/s (dose and route if specified) | Results | Dominance/cost-effectiveness | Summary of study conclusions |
|---|---|---|---|---|
| Khan and colleagues [83] | 1. CS with cell salvage<br>2. CS without cell salvage | • Standard care was least costly with an estimated average cost per patient of £1,244 GBP 2014 ($2,104 USD 2023).<br>• The cell salvage group was slightly more expensive, with the average cost per patient estimated at £1,327 ($2,244 USD 2023).<br>• The average incremental cost incurred was £8,110 ($13,714 USD 2023) per donor blood transfusion avoided by utilizing cell salvage. | No threshold set. | Although cell salvage is marginally more effective than standard care in avoiding donor blood transfusions, it is unlikely that cell salvage would be considered cost-effective. |
| Lim and colleagues [84] | 1. Cell salvage for all CS.<br>2. Cell salvage only for deliveries at high risk for haemorrhage (including placenta previa, placenta accreta, repeat CS or multiparity, chorioamnionitis, placental abruption, hypertensive disorders during pregnancy, etc.).<br>3. No utilisation of cell salvage. | Base case:<br>• The incremental cost incurred for using cell salvage on high-risk CS cases was $34,881 USD 2012 ($42,546 USD 2023) per QALY gained.<br>• The incremental cost incurred for using cell salvage on all CS cases was $415,488 ($506,787 USD 2023) per QALY gained.<br><br>OWSA:<br>• Results were not sensitive to individual variation of other model parameters.<br><br>PSA:<br>• At the $100,000 ($121,974 USD 2023) per QALY gained threshold, there is more than 85% likelihood that cell salvage use for cases at high risk for haemorrhage is cost-effective. | Cell salvage for high-risk CS: cost-effective at author's stated threshold of $100,000 ($121,974 USD 2023) per QALY gained. Cell salvage for all CS: Not cost-effective. | Cell salvage for cases at high risk for haemorrhage is economically reasonable compared to strategies of cell salvage use for all CS or no cell salvage use at all. |
| Ries and colleagues [88] | 1. Implementation of the "D-A-CH Handlungsalgorithmus Postpartale Blutung" algorithm for the management of PPH.<br>2. PPH management prior to the implementation of the D-A-C-H algorithm. | • There was no significant difference in clinical outcomes (estimated blood loss, ICU transfers, and Hb value 2 days postpartum).<br>• Implementation of the algorithm did however result in the usage of a wider range of pharmacological interventions for PPH treatment within a shorter time interval after delivery.<br>• Treatment in the intervention group was not statistically more expensive on average than before the implementation of the algorithm: control ₣434.7, vs. intervention ₣233.9 CHF, year not stated, $p = 0.571$ ($377.16 vs. $202.94 USD 2023).<br><br>First year of data collection (2009) used for cost conversion calculation. | No threshold set. | Implementation of the treatment algorithm in women after VB with severe PPH did not result in significantly reduced blood loss. Implementation did however accelerate clinical management and induced the application of a wider range of pharmacological interventions closer to the time of delivery and did not generate more costs. |

*(Continued)*

**Table 4.** (Continued)

| Study | Intervention/s and comparator/s (dose and route if specified) | Results | Dominance/cost-effectiveness | Summary of study conclusions |
|---|---|---|---|---|
| Franke and colleagues [89] | 1. Referral and transport to secondary care facility<br>2. No referral (primary care only) | • 46 women used the referral and transport system due to PPH.<br>• The intervention was estimated to incur $17.10, USD 2020 ($19.82 USD 2023) per life year saved for PPH patients. | No threshold stated. | The intervention was found to be very cost-effective and may assist with public health resource allocation in Madagascar. |

Economic results are stated as they appear in the original publications and as a conversion to USD 2023 using an online tool developed by the Campbell and Cochrane Economics Methods Group (CCEMG) and the Evidence for Policy and Practice Information and Coordinating Centre (EPPI-Centre) [19]. Cost conversions were completed in December 2023 and may change slightly depending on final GDP figures.

CHF, Swiss Franc; CS, cesarean section; DALYs, disability-adjusted life years; ESM-UBT, every second matters—uterine balloon tamponade; EUR, Euro; FFP, fresh frozen plasma; GBP, British Pound; GDP, gross domestic product; ICER, incremental cost-effectiveness ratio; ICU, intensive care unit; INR, Indian rupee; IV, intravenous; MAP, mean arterial pressure; Hb, haemoglobin; NASG, non-pneumatic anti-shock garment; NHS, National Health Service; OWSA, one-way sensitivity analysis; PCVT, point of care viscoelastic testing; PO, per oral; PPH, postpartum haemorrhage; PR, per rectum; PRBC, packed red blood cells; PSA, probabilistic sensitivity analysis; QALYs, quality-adjusted life years; RBC, red blood cells; TBA, traditional birth attendant; TXA: tranexamic acid; UBT, uterine balloon tamponade; VB, vaginal birth; μg, microgram.

authors could not conclude that switching from improvised devices to ESM-UBT in India would be cost-effective [78]. The assessment of one further device was identified, the novel Butterfly device for uterine compression [80]. This device is not yet utilised outside of trials and the results are reported in Table 3.

**Use of blood products for PPH management.** Five studies assessed various strategies for optimising blood product management in women with PPH [81–85]. The first, an economic evaluation from trial data, assessed blood resuscitation for severe PPH guided by point-of-care viscoelastic testing (PCVT) compared to empiric blood product resuscitation in the USA [82]. The intervention group experienced lower volumes of estimated blood loss, lower hysterectomy and intensive care unit admission rates, and significantly lower hospital costs [82]. However, the study was low quality, may be at risk of bias related to treatment effect (Table U in S8 Appendix) and it is unclear if the device-associated costs were included in the analysis.

A high-quality modelling study in the USA attempted to identify the most cost-effective strategy for predelivery blood type, screening, and cross-matching [81]. Although some strategies were successful at reducing emergency-release transfusions, the upfront costs of predelivery testing outweighed the benefits; the best strategy was no routine admission testing [81]. Possible bias related to treatment effects could not be assessed (Table U in S8 Appendix).

One trial-based study in the Netherlands assessed whether women with acute anaemia following PPH should be transfused to a target haemoglobin or treated conservatively with iron and folic acid supplementation [85]. Although no clear cost-effectiveness threshold was set, the large costs required to lift women's fatigue and quality of life scores by marginal amounts were not justified from the hospital perspective [85].

Two modelling studies assessing cell salvage during cesarean section were identified [83,84]. In the UK study, cell salvage in cesarean section compared to no cell salvage incurred $13,714 per donor blood transfusion avoided, and was therefore unlikely to be cost-effective [83]. In the USA study, a broader study perspective, longer time horizon, and higher effectiveness estimates were used [84]. The authors concluded that cell salvage when women were at high risk of haemorrhage was likely cost-effective at their stated thresholds [84]. Both studies were high quality, though possible bias related to treatment effects could not be assessed due to the considerable variability of published effectiveness estimates [86,87].

**Treatment algorithms for PPH management.** One study evaluated a PPH treatment algorithm in Switzerland [88]. Introducing the algorithm resulted in a wider range of PPH treatments being applied in more rapid succession, without showing statistically significant differences in cost or health outcomes [88].

**Referral to higher level care for PPH management.** One high-quality modelling study assessed the cost-effectiveness of an emergency interfacility referral and transfer service for women and neonates in rural Madagascar [89]. The service, which assessed and transported patients from primary level care to secondary level facilities was estimated to be highly cost-effective, incurring $19.82 per additional life year saved for women with PPH. However, these results should be interpreted cautiously as the sub-analysis of women with PPH included only 46 patients, and survival rates were based on expert opinion rather than primary data.

### Economic evaluations of PPH bundles

Care bundles are a complex strategy where multiple interventions are used simultaneously [90]. Three evaluations using effectiveness trial data [91–93], and 1 modelling study [94] considered care bundles for PPH-related care (Table 5). Two evaluated a combination of preventative and treatment interventions [92,94]. A study in Niger reported a bundle of preventative uterotonics, improvised blood loss measurement and tiered treatment responses was highly effective and incurred low incremental costs [92]. A USA study showed that PPH preparedness and treatment strategies could be bundled, improving maternal health outcomes and reducing costs [94].

Two studies evaluated bundles combining early detection and treatment interventions [91,93]. A study in Wales reported that their bundle—comprising universal risk assessments for PPH, early PPH identification through quantitative measurement, mandatory multidisciplinary team involvement at certain blood loss thresholds, and point-of-care coagulation testing after 1,000 ml blood loss—modestly reduced the proportion of PPH cases progressing to massive PPH, at a cost of $29.16 per patient with PPH > 1,000 ml [91]. The second study, a health economic evaluation of the E-MOTIVE trial in four sub-Saharan African countries, reported their intervention was highly cost-effective, incurring only $118.10 per DALY averted [93]. CHEC-E scores and certainty of effectiveness data of the 4 bundle evaluations varied (Table D in S7 Appendix and Table V in S8 Appendix).

## Discussion

This is, to our knowledge, the first systematic review examining the cost-effectiveness of interventions for PPH across the continuum of prevention, diagnosis, treatment, or combinations of these. We identified 56 studies—approximately half (24 studies) were conducted in high-income settings. Despite this considerable body of economic evidence for PPH-related care, the interventions, evaluation methodologies, time horizons, and perspectives varied considerably between studies. Acknowledging this heterogeneity, some patterns emerged. Currently, no injectable uterotonic agent or combination is universally dominant from a cost-effectiveness perspective. Across 15 studies assessing carbetocin against oxytocin, studies were split as to whether carbetocin improved health outcomes while incurring further costs or saved costs. The single UK study that considered the full range of available uterotonics did not recommend switching away from oxytocin [30,31].

Six studies found that when injectable uterotonics are unavailable using misoprostol is consistently cost-effective over no uterotonic [52–57]. However, the quality of studies was variable, and effectiveness estimates underpinning the cost-effectiveness calculations are inconsistent

**Table 5. Results of bundle intervention studies.**

| Study | Intervention/s and comparator/s (dose and route if specified) | Results | Dominance/cost-effectiveness | Summary of study conclusions |
|---|---|---|---|---|
| Seim and colleagues [92] | 1. Standard care, plus: health worker education, misoprostol distribution for home deliveries, oxytocin for use in hospitals/health centres, semi-quantitative blood loss measurement, 3 step treatment for PPH with additional uterotonics, intrauterine condom tamponade, NASG and transfer for definitive management.<br>2. Standard care in Niger prior to the above nationwide intervention being launched. | • Case fatality rate per 100 primary PPH cases decreased from 5.05% (95% CI 3.36–7.30) to 2.58% (2.18–3.03%).<br>• The intervention was estimated to cost $27.73–$37.94 USD 2013 ($33.24–45.48 USD 2023) per DALY averted. | No threshold specified. | The low-cost intervention more than halved the mortality associated with primary PPH within 2 years of implementation. |
| Wiesehan and colleagues [94] | 1. California's statewide perinatal quality collaborative initiative, including: haemorrhage cart, rapid access to PPH medications, dedicated haemorrhage response team, massive transfusion protocol, staff training and drills, haemorrhage risk assessments, department wide active management of third stage of labour with oxytocin, and many other elements (complete list in Table 1 of this study).<br>2. Standard care in Californian hospitals not taking part in the quality collaborative. | Base case<br>• The intervention was modelled to increase QALYs by 0.000379 and reduce costs by $17.78 USD 2021 ($18.52 USD 2023) per birth.<br>OWSA:<br>• Cost-effective in every analysis (at author's stated threshold of $100,000 ($104,184 USD 2023) per QALY gained).<br>PSA:<br>• Cost saving in 83% of samples and cost-effective in 99% of samples (at stated threshold). | Bundle intervention dominant compared to standard care. | The intervention is inexpensive, reduces severe maternal morbidity and mortality and is potentially cost saving. |
| Dale and colleagues [91] | 1. Standard care, plus universal risk assessments for PPH, quantitative blood loss measurement, multidisciplinary team management of PPH, point of care coagulation blood testing after 1,000 ml blood loss to guide resuscitation.<br>2. Standard care in Wales prior to the above nationwide process improvements. | • No difference in the number of cases of PPH > 1,000 ml.<br>• Reduction in the number of cases progressing from PPH > 1,000 ml to PPH > 2,500 ml from 6.9 per 1,000 births to 5.2 per 1,000 births.<br>• Decrease in resource use for blood products, critical care, and haematologist time.<br>• Incremental cost of £18.41 GBP 2018 ($29.16 USD 2023) per patient with PPH > 1,000 ml. | No threshold specified. | The intervention reduced the occurrence of massive PPH, the quantity of blood products used, and intensive care resources utilised. The incremental costs incurred decrease towards cost-neutrality in medium-large (>3,000 annual births) maternity units. |
| Williams and colleagues [93] | 1. Early PPH detection and treatment with: quantitative blood loss measurement, uterine massage, oxytocin, TXA, IV fluids, examination, and escalation to definitive management if needed, staff training and auditing, dedicated PPH-response trolley/case.<br>2. Standard care in the respective hospitals and countries. | • The intervention bundle would incur costs of $11.83 USD 2022 ($12.26 USD 2023) for each case of severe PPH averted or $113.91 ($118.10) for each DALY averted. | Cost-effective at threshold based on GDP per capita of $2,816 per DALY averted ($2,920 USD 2023).<br>Also, cost-effective at threshold based on opportunity cost of $1,690 per DALY ($1,752 USD 2023). | Early detection of PPH using a calibrated blood-loss collection drape and treatment with the WHO first-response bundle is cost-effective compared with usual care. |

Economic results are stated as they appear in the original publications and as a conversion to USD 2023 using an online tool developed by the Campbell and Cochrane Economics Methods Group (CCEMG) and the Evidence for Policy and Practice Information and Coordinating Centre (EPPI-Centre) [19]. Cost conversions were completed in December 2023 and may change slightly depending on final GDP figures.

DALYs, disability-adjusted life years; GBP, British Pound; GDP, gross domestic product; IV, intravenous; NASG, non-pneumatic anti-shock garment; OWSA, one-way sensitivity analysis; PPH, postpartum haemorrhage; PSA, probabilistic sensitivity analysis; QALYs, quality-adjusted life years; TXA, tranexamic acid; USD, United States Dollar.

with recent estimates [30,37]. Studies of tranexamic acid for either prevention or treatment scored high on CHEC-E, used appropriate effectiveness estimates and concluded that the addition of tranexamic acid either dominated or was cost-effective across a range of settings [59–61,72–75], acknowledging that in some circumstances, the magnitude of the health benefit was small [61]. The 4 studies assessing the combination of multiple interventions into care bundles —variably involving prevention, diagnosis, and treatment components—suggest this can provide good economic value [91–94], particularly in LMICs [92,93].

Reviewing the effectiveness data used in each study's cost-effectiveness calculations provided valuable insights. In some economic evaluations, the magnitude of an intervention's effect was dramatically different to effect estimates reported in reputable systematic reviews. For example, one study's base-case evaluation was centred around 600 µg of oral misoprostol being more than 6 times more effective at preventing transfer to hospital due to PPH, than 10 IU (international units) of intramuscular oxytocin for women in maternity huts in Senegal [47]. The trial this estimate was based on only had a single event for this outcome [95]. In other studies, assessing the cost-effectiveness of misoprostol administered by skilled staff, misoprostol was repeatedly assumed to lower the risk of PPH by 50% to 80% compared to no medication [52,56,57]. More recent effect estimates from systematic reviews indicate that it likely reduces the risk of PPH by approximately 25% [37]. The difference in assumed effectiveness can greatly change the number of expected outcomes, and therefore how cost-effective the intervention is perceived to be. For illustration, in a cohort of 10,000 women giving birth, with a baseline rate of PPH of 6%, altering the assumed effectiveness of misoprostol from reducing risk of PPH by 50% down to 25% would result in an extra 150 cases of PPH, and considerably change the costs used in the evaluation.

Previous systematic reviews have focused only on the cost-effectiveness of uterotonics for prevention [11], intrauterine devices for treatment [12], or tranexamic acid for treatment [13]. Similar to the 2019 review of preventative uterotonics, we were unable to clearly identify a particular agent that performed better from an economic perspective, despite a doubling of included studies. In comparison with the 2020 review on uterine tamponade devices, we identified one further economic evaluation [78], though economic data on this intervention remains sparse. Since the 2021 review on tranexamic acid for PPH treatment, one further economic evaluation has been published [74], and its results are compatible with the previous conclusion—that it is likely either cost-saving or cost-effective.

Although there are robust, comparative effectiveness data for the use of injectable uterotonics in PPH prevention [37], there remains no universally dominant choice from an economic perspective. The most economical choice of uterotonic is context-specific—considering trade-offs in effectiveness, side-effect profile, cost, availability, heat-stability, and the presence of trained providers to administer them. By comparison, the economic rationale for incorporating tranexamic acid into PPH treatment is more straightforward. It is recommended by WHO [23,29] with compelling evidence of benefits [96], and it has been demonstrated to be cost-saving or cost-effective in diverse settings [72–75]. In contrast, tranexamic acid for PPH prevention is not currently recommended by WHO. Effectiveness evidence of its prophylactic use suggests some benefit in preventing postpartum blood loss [7,62,63,97], noting that the 3 economic evaluations suggesting that it is a dominant or cost-effective strategy were from high-income settings [59–61]. While the addition of single agents to PPH prevention or treatment protocols can offer clinical and economic benefits, the implementation of combining therapies using a care bundle approach offers another path forward. There is high-quality economic evidence that care bundles can reduce the burden of PPH in LMIC [90,98] at modest incremental costs [93].

This review identified large gaps in the health economic literature that require further research. Several pivotal PPH interventions—such as oxytocin for treating PPH and quantitative measurement of postpartum blood loss—do not have high-quality economic data. Other interventions, such as PPH prevention using misoprostol have numerous economic evaluations, yet all were based on effectiveness estimates that have been superseded and are therefore of limited utility. Some interventions such as UBT and NASG demonstrated good value economically but have only been evaluated in a small number of studies and settings.

While effectiveness evidence forms the basis for recommendations, cost-effectiveness and resource use data play a key role in their implementability, especially so for resource-limited settings. Thus, effectiveness trials on PPH prevention, diagnosis, and management should have economic evaluations routinely embedded, helping maximise their translation and impact. The E-MOTIVE trial [98], and its embedded economic evaluation [93,99] is a recent successful example. First-order approximations of how results could translate across settings are possible to an extent with modelling that accounts for different demographics, health system characteristics, baseline intervention coverage and costs. These can be used to inform local, context-specific policies, but should not be used in the absence of considering local acceptability and feasibility. Translatability of results from modelling and our ability to make evidence informed decisions based on their results would also benefit from the creation of more robust methods for evaluating the certainty of economic evaluations [100].

Strengths of this review include the broad search conducted on multiple databases, with no limitations on language or date, and the systematic methodology employed in screening studies, extracting data, appraising quality, and resolving disputes. In addition, only peer-reviewed, economic analyses including the assessment of both health and financial outcomes were included in this review. Beyond appraising study quality, we also compared the effectiveness data used in each study to up-to-date systematic reviews, to minimise the possibility that economic findings based on outdated effectiveness data had influenced our conclusions.

This review has several limitations. Firstly, it was completed without nominating an ideal setting. Without an ideal setting, no transferability assessment could be completed. It is therefore important that users of the collated and summarised evidence in this review consider the extent to which each study finding is applicable to their own (or a nominated) setting. Secondly, while some major sources of study uncertainty were interrogated through the CHEC-E assessment, the description of differences between study settings and methods, and the in-depth assessment of effectiveness parameters used, other possible sources of uncertainty were not able to be interrogated to the same extent. In particular, a summary of the costing parameters used, and model validation methods undertaken by each study were not completed beyond the requirements of the CHEC-E assessment tool. Thirdly, our review of effectiveness estimates relied on recent systematic reviews; more recent data from single randomised trials, not yet incorporated into meta-analyses, are thus not captured. Fourthly, by choosing to summarise economic evaluations across prevention, diagnosis and treatment of PPH this review has prioritised breadth over the depth of analysis in the main manuscript. More detail, such as an analysis of the most appropriate study perspective and time horizons would have been possible through single intervention reviews on a smaller group of studies. It is important that the results summarised in this review are interpreted in the context of the more detailed information contained in the supplementary materials, namely the study characteristics (S6 Appendix), CHEC-E scores (S7 Appendix), and the analysis of the effectiveness data used in each study (S8 Appendix). Fifthly, several included studies were relatively old—7 were published prior to 2014. While historical cost estimates can be adjusted to present value, the treatment choices, evidence to inform key model parameters, and methodologies used to conduct and report economic evaluations have evolved over time.

## Conclusion

This review has collected, summarised, and highlighted important health economic findings from 56 studies across 16 interventions for the prevention, diagnosis, and treatment of PPH. We identified consistent evidence that adding tranexamic acid to PPH treatment regimens is a dominant strategy and that combining PPH interventions into bundles can deliver improved health outcomes for modest cost. We also identified significant gaps in the cost-effectiveness evidence for PPH interventions. Of the 29 WHO recommendations, 16 do not have any cost-effectiveness evidence, and further assessments of widely used interventions for PPH treatment such as additional uterotonics, non-pneumatic anti-shock garment, or uterine balloon tamponade are urgently needed.

## Supporting information

**S1 Appendix. Prisma checklist.**
(DOCX)

**S2 Appendix. Search strategies.**
(DOCX)

**S3 Appendix. Data extraction variables.**
(DOCX)

**S4 Appendix. CHEC-Extended questions.**
(DOCX)

**S5 Appendix. Excluded studies.**
(DOCX)

**S6 Appendix. Characteristics of included studies.**
(DOCX)

**S7 Appendix. CHEC-Extended scores.**
(DOCX)

**S8 Appendix. Analysis of effectiveness data used in included studies.**
(DOCX)

**S9 Appendix. Economic evaluations mapped to WHO recommendations.**
(DOCX)

**S10 Appendix. Results presented by region.**
(DOCX)

**S11 Appendix. Summary of economic evaluations of oxytocin for prevention.**
(DOCX)

**S12 Appendix. Summary of economic evaluations of surgical interventions.**
(DOCX)

## Acknowledgments

Lorena Romero MBIT, Research & Training Librarian, The Ian Potter Library, Alfred Health, Melbourne, Victoria, Australia.

Jiaxin Huang, MPH Student, University of Melbourne, Victoria, Australia.

## Author Contributions

**Conceptualization:** Joshua F. Ginnane, Annie McDougall, Katherine E. Eddy, Joshua P. Vogel.

**Data curation:** Joshua F. Ginnane, Samia Aziz, Saima Sultana, Connor Luke Allen, Annie McDougall, Katherine E. Eddy.

**Formal analysis:** Joshua F. Ginnane, Samia Aziz, Saima Sultana.

**Project administration:** Joshua F. Ginnane, Joshua P. Vogel.

**Supervision:** Joshua P. Vogel.

**Visualization:** Joshua F. Ginnane.

**Writing – original draft:** Joshua F. Ginnane.

**Writing – review & editing:** Joshua F. Ginnane, Annie McDougall, Katherine E. Eddy, Nick Scott, Joshua P. Vogel.

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
