## [Editor Report · Decision Letter 0]

27 Apr 2024

Dear Dr Ginnane, 

Thank you for submitting your manuscript entitled "The cost-effectiveness of preventing, diagnosing and treating postpartum haemorrhage: a systematic review of economic evaluations" for consideration by PLOS Medicine.

Your manuscript has now been evaluated by the PLOS Medicine editorial staff and I am writing to let you know that we would like to send your submission out for external peer review.

Please re-submit your manuscript within two working days, i.e. by May 01 2024 11:59PM.

Feel free to email me at lgaynor@plos.org if you have any queries relating to your submission.

Kind regards,

Louise Gaynor-Brook, MBBS PhD

Senior Editor

PLOS Medicine

---

## [Decision Letter · Decision Letter 1]

28 May 2024

Dear Dr. Ginnane,

Thank you very much for submitting your manuscript "The cost-effectiveness of preventing, diagnosing and treating postpartum haemorrhage: a systematic review of economic evaluations" (PMEDICINE-D-24-01325R1) for consideration at PLOS Medicine. 

Your paper was evaluated by three independent reviewers, including a statistical reviewer, and discussed among all the editors here and with an academic editor with relevant expertise. The reviews are appended at the bottom of this email and any accompanying reviewer attachments can be seen via the link below:

[LINK]

I’m pleased to invite you to revise the paper in response to the reviewers’ and editors' comments below. Obviously we cannot make any decision about publication until we have seen the revised manuscript and your response, and we plan to seek re-review by one or more of the reviewers. 

We expect to receive your revised manuscript by Jun 18 2024 11:59PM. Please email me (lgaynor@plos.org) if you have any questions or concerns.

We look forward to receiving your revised manuscript. 

Sincerely,

Louise Gaynor-Brook, MBBS PhD

PLOS Medicine

plosmedicine.org

lgaynor@plos.org

We would like to see a more thorough discussion of limitations in the Discussion section. 

In agreement with reviewer 3, we would also be interested in seeing results presented by region in the supplementary files, if this is feasible. 

Comments from the reviewers:

Reviewer #1: a) The manuscript mentions that for each study, effectiveness data were extracted and compared with up-to-date estimates from corresponding WHO recommendations and systematic reviews. It would be beneficial for the reader to understand the impact of these up-to-date effectiveness estimates on the overall cost-effectiveness results. Could you please provide examples or a detailed discussion on how significant changes in effectiveness estimates have influenced the economic outcomes in specific studies?

b) The paper states that a transferability assessment was not completed due to the absence of an ideal setting for comparison. However, understanding the applicability of these economic evaluations in different healthcare contexts is crucial. Could the authors discuss the potential limitations this poses on the generalizability of the study findings? 

c) Regarding the mapping of studies to WHO recommendations, the methodology used for this alignment is not clearly outlined. Clarifying how studies were associated with specific WHO recommendations would enhance the transparency of the process. Additionally, it would be insightful if the authors could discuss whether the findings suggest any modifications or reaffirmations to existing WHO recommendations based on the newly reviewed economic data.

d) The manuscript could benefit from a more detailed exploration of uncertainties associated with the studies reviewed, particularly how these uncertainties impact the cost-effectiveness of interventions. Were there any specific resource use parameters within these studies that were identified as highly uncertain? How might these uncertainties affect the reliability of the cost-effectiveness results?

e) For the economic evaluations that utilized modelling approaches, the manuscript does not provide sufficient details on the validity and reliability of these models. Could the authors include a section discussing the assumptions made within these models, the sources of data used for parameter estimates, and any validation processes undertaken? 

f) While the manuscript reviews various economic evaluations, it does not provide a detailed critique of the methodologies used across these studies. A deeper examination of the strengths and limitations of different economic evaluation methods would be beneficial.

Reviewer #2: Introduction

1. While the health outcomes associated with PPH are made clear, it would be useful to provide an indication of the associated economic burden. Incorporating this would provide readers with a more comprehensive understanding of the overall impact of PPH. 

2. I feel that further justification is needed to demonstrate the relevance of reviewing evidence from high-income countries, particularly when the authors highlight that deaths from PPH have largely been eliminated in HICs, that PPH disproportionately affects patients in LMICs, and that results indicate that cost-effectiveness of such interventions can be highly contextual. 

3. Cost-effectiveness evidence relating to many interventions for prevention, detection and treatment of PPH is summarised in the paper. Including an introductory section with more detail on these interventions could be useful.

Methods

1. More clarity is needed on the eligibility criteria. The authors state that full economic evaluations were included, whilst partial economic evaluations were excluded. However, a number of cost-consequence analyses (which do not permit full incremental analyses) are included. A longitudinal study is also included, which is not a full economic evaluation.

2. More information on the method used to convert costs to 2023 USD would be useful. Was this using exchange rates or a PPP adjustment approach?

Results

1. It would be useful to summarise more characteristics of the included studies, such as setting (community, hospital), economic evaluation type (model vs trial-based analysis) and perspective (societal, healthcare payer).

2. Many studies assessed the cost-effectiveness of prophylactic uterotonics, and consequently that sub-section of the results is quite long. Could it be subdivided further to make it a little easier to digest?

3. It was not clear why the studies assessing oxytocin to no uterotonic, non-injectable uterotonics or other formulations of oxytocin were not detailed in the main text whilst the single study on AMTSL was. Similarly, it was not overly clear why the studies on surgical methods for minimising PPH were not included in the main text. Was it due to their low quality?

4. At times I felt that more context is needed when summarising the cost-effectiveness findings. The authors acknowledge the considerable heterogeneity between studies in the discussion, but it doesn't really come across in the results section. Including a more critical/comprehensive overview of key study aspects in the study results could enhance understanding and facilitate a more nuanced interpretation of the findings. Though, I appreciate this can be tricky when so many studies are included!

Discussion

1. The summary of findings provided in the discussion is appropriate, however it could be more succinct. 

2. The conclusion presented in the discussion should also include the detail included in the conclusion of the abstract.

Reviewer #3: Thank you for allowing me to read this well conducted systematic review of economic evaluations of diagnostic and therapeutic interventions for post partum haemorrhage. The methods are fully described with the checklists used and selection criteria. The choice of including interventions in all countries makes the article relevant for all readers. 

The drawback is that the presentation of the results is difficult, I suppose the authors have struggled to decide whether to present by country/region or, as they did, by intervention. I would suggest to add the regional aspect in the supplemental material (with a table, either a new one by region, or colour code the existing one to allow easy identification of the setting) and in the Discussion because interventions that are highly feasible in a high income setting (eg an iv line for TXA) might be more difficult in other settings. I would be more prudent about modeling to translate results across settings (p12), because this is limited by the availability of resources.

* Please note that not all of these will be applicable to your manuscript. * 

Please ensure that the paper adheres to the PLOS Data Availability Policy (see http://journals.plos.org/plosmedicine/s/data-availability), which requires that all data underlying the study's findings be provided in a repository or as Supporting Information. For data residing with a third party, authors are required to provide instructions with contact information (web or email address) for obtaining the data. Please note that a study author cannot be the contact person for the data. PLOS journals do not allow statements supported by "data not shown" or "unpublished results." For such statements, authors must provide supporting data or cite public sources that include it. 

We expect all researchers with submissions to PLOS in which author-generated code underpins the findings in the manuscript to make all author-generated code available without restrictions upon publication of the work. In cases where code is central to the manuscript, we may require the code to be made available as a condition of publication. Authors are responsible for ensuring that the code is reusable and well documented. Please make any custom code available, either as part of your data deposition or as a supplementary file. Please add a sentence to your data availability statement regarding any code used in the study; eg “The code used in the analysis is available from Github [URL] and archived in Zenodo [DOI link]” Please review our guidelines at h https://journals.plos.org/plosmedicine/s/materials-software-and-code-sharing and ensure that your code is shared in a way that follows best practice and facilitates reproducibility and reuse. 

We ask every co-author listed on the manuscript to fill in a contributing author statement, making sure to declare all competing interests. If any of the co-authors have not filled in the statement, we will remind them to do so when the paper is revised. If all statements are not completed in a timely fashion this could hold up the re-review process. If new competing interests are declared later in the revision process, this may also hold up the submission. Should there be a problem getting one of your co-authors to fill in a statement we will be in contact. Please do not add or remove authors without first discussing this with the handling editor. You can see our competing interests policy here: http://journals.plos.org/plosmedicine/s/competing-interests. 

Please upload any figures associated with your paper as individual TIF or EPS files with 300dpi resolution at resubmission; please read our figure guidelines for more information on our requirements: http://journals.plos.org/plosmedicine/s/figures. While revising your submission, please upload your figure files to the PACE digital diagnostic tool, https://pacev2.apexcovantage.com/. PACE helps ensure that figures meet PLOS requirements. To use PACE, you must first register as a user. Then, login and navigate to the UPLOAD tab, where you will find detailed instructions on how to use the tool. If you encounter any issues or have any questions when using PACE, please email us at PLOSMedicine@plos.org. 

FORMATTING 

Abstract: Please structure your abstract using the PLOS Medicine headings (Background, Methods and Findings, Conclusions). Please combine the Methods and Findings sections into one section, “Methods and findings”. 

At this stage, we ask that you include a short, non-technical Author Summary of your research to make findings accessible to a wide audience that includes both scientists and non-scientists. The Author Summary should immediately follow the Abstract in your revised manuscript. This text is subject to editorial change and should be distinct from the scientific abstract. Ideally each sub-heading should contain 2-3 single sentence, concise bullet points containing the most salient points from your study. In the final bullet point of ‘What Do These Findings Mean?’, please include the main limitations of the study in non-technical language. Please see our author guidelines for more information: https://journals.plos.org/plosmedicine/s/revising-your-manuscript#loc-author-summary. 

Please express the main results with 95% CIs as well as p values. When reporting p values please report as p<0.001 and where higher as the exact p value p=0.002, for example. Throughout, suggest reporting statistical information as follows to improve clarity for the reader “22% (95% CI [13%,28%]; p</=)”. Please be sure to define all numerical values at first use. 

Please include page numbers and line numbers in the manuscript file. Use continuous line numbers (do not restart the numbering on each page). 

Please cite the reference numbers in square brackets. Citations should precede punctuation. 

FIGURES AND TABLES 

Please provide titles and legends for all figures and tables (including those in Supporting Information files). 

Please define all abbreviations used in each figure/table (including those in Supporting Information files). 

Please consider avoiding the use of red and green in order to make your figure more accessible to those with color blindness. 

SUPPLEMENTARY MATERIAL 

Please note that supplementary material will be posted as supplied by the authors. Therefore, please amend it according to the relevant comments. 

Please cite your Supporting Information as outlined here: https://journals.plos.org/plosmedicine/s/supporting-information

REFERENCES 

PLOS uses the numbered citation (citation-sequence) method and first six authors, et al. 

Please ensure that journal name abbreviations match those found in the National Center for Biotechnology Information (NCBI) databases (http://www.ncbi.nlm.nih.gov/nlmcatalog/journals), and are appropriately formatted and capitalised. 

Where website addresses are cited, please include the complete URL and specify the date of access (e.g. [accessed: 12/06/2023]). 

Please also see https://journals.plos.org/plosmedicine/s/submission-guidelines#loc-references for further details on reference formatting. 

SYSTEMATIC REVIEWS & META-ANALYSES 

Please report your SR/MA according to the PRISMA guidelines provided at the EQUATOR site. http://www.equator-network.org/reporting-guidelines/prisma/. Please provide the completed PRISMA checklist as Supporting Information. When completing the checklist, please use section and paragraph numbers, rather than page numbers. 

Please add the following statement, or similar, to the Methods: "This study is reported as per the Preferred Reporting Items for Systematic Reviews and Meta-Analyses (PRISMA) guideline (S1 Checklist)." 

Abstract: Please report your abstract according to PRISMA for abstracts (https://doi.org/10.1371/journal.pmed.1001419) following the PLOS Medicine abstract structure (Background, Methods and Findings, Conclusions). Please ensure you provide dates of search, data sources, number of studies included, types of study designs included, eligibility criteria, and synthesis/appraisal methods. 

[LINK]

---

## [Decision Letter · Decision Letter 2]

10 Jul 2024

Dear Dr. Ginnane,

Thank you very much for re-submitting your manuscript "The cost-effectiveness of preventing, diagnosing and treating postpartum haemorrhage: a systematic review of economic evaluations" (PMEDICINE-D-24-01325R2) for review by PLOS Medicine.

Thank you for your detailed response to the editors' and reviewers' comments. I have discussed the paper with my colleagues, and it has also been seen again by two of the original reviewers. The changes made to the paper were mostly satisfactory to the reviewers. As such, we intend to accept the paper for publication, pending your attention to the remaining reviewer and editorial comments below in a further revision. When submitting your revised paper, please once again include a detailed point-by-point response to the editorial comments.

[LINK]

In revising the manuscript for further consideration here, please ensure you address the specific points made by each reviewer and the editors. In your rebuttal letter you should indicate your response to the reviewers' and editors' comments and the changes you have made in the manuscript. Please submit a clean version of the paper as the main article file. A version with changes marked must also be uploaded as a marked up manuscript file. Please also check the guidelines for revised papers at http://journals.plos.org/plosmedicine/s/revising-your-manuscript for any that apply to your paper.

We ask that you submit your revision within 1 week (Jul 17 2024). However, if this deadline is not feasible, please contact me by email, and we can discuss a suitable alternative.

Please do not hesitate to contact me directly with any questions (lgaynor@plos.org). If you reply directly to this message, please be sure to 'Reply All' so your message comes directly to my inbox.

We look forward to receiving the revised manuscript.   

Sincerely,

Alexandra Tosun, PhD

On behalf of

Louise Gaynor-Brook, MBBS PhD

Senior Editor 

PLOS Medicine

plosmedicine.org

Requests from Editors:

We require that systematic reviews are updated to within roughly 6 months of the expected publication date. Please update your search to the present time.

General comments:

1) Throughout the paper, please adapt reference call-outs to the following style: "... every year [1,2]." (noting the absence of spaces within the square brackets, and placement before the full stop).

2) Please include page numbers and line numbers in the manuscript file. Use continuous line numbers (do not restart the numbering on each page).

3) To help us extend the reach of your research, please provide any X (formerly known as Twitter) handle(s) that would be appropriate to tag, including your own, your co-authors’, your institution, funder, or lab. Please enter in the submission form any handles you wish to be included when we post about this paper.

ABSTRACT

1) Please provide the beginning and end dates of your search.

2) In the last sentence of the Abstract Methods and Findings section, please describe 2-3 of the main limitations of the study's methodology.

3) Please begin your Abstract Conclusions with "In this study, we..." or similar, to summarize the main findings from your study, without overstating your conclusions. Please emphasize what is new and address the implications of your study, being careful to avoid assertions of primacy. 

4) Please define WHO at first use.

AUTHOR SUMMARY

1) Thank you for providing an Author Summary. This should be formatted as 2-3 single sentence bullet points for each of the questions. Bullet points should be objective, brief, succinct, specific, accurate, and avoid technical language.

2) In the final bullet point of ‘What Do These Findings Mean?’, please describe the main limitations of the study in non-technical language.

INTRODUCTION

If there has been a previous systematic review of the evidence related to your study (or you have conducted one), please refer to and reference that review and indicate whether it supports the need for your study. 

METHODS AND RESULTS

1) Thank you for providing the completed PRISMA checklist. Please revise the checklist and use section and paragraph numbers, rather than page numbers which will likely no longer correspond to the appropriate sections after copy-editing.

2) As stated above, please update your search to the present time and provide the beginning and end dates of your search.

3) Please report the total number of WHO recommendations (under "Characteristics of included studies"; for comparison purposes).

4) Figure 1: Thank you for providing the flow diagram. Please remove the reference from the figure description. It is sufficient to provide the reference in the appendix, as you have already done. There are two asterisks next to "Records excluded" in the figure without any definition in the description, please revise. We also suggest replacing the word "wrong" in the figure with "other" or similar to avoid any negative implications. 

5) Figure 2: Please define IV, PPH, WHO. Please mention the total number of studies identified in the systematic review in the figure description. Also, please add more details about what the top and bottom halves of the figure represent, e.g. “The top half shows the WHO-recommended PPH interventions at different stages postpartum, while the bottom half shows the interventions identified in the economic evaluation studies identified in this systematic review.”.

6) Please reduce the number of subheadings within the Results section. For example, "Prophylactic uterotonics" should only be used as one subheading, with comprehensive paragraph introductions guiding the reader through the sections (e.g. “For evaluations considering carbetocin as the primary intervention, five studies used decision analytic models to compare carbetocin with oxytocin for both vaginal birth and caesarean section [31-35].”).

7) Please include numerators and denominators when describing the number of studies identified for a particular intervention. For example, "Twenty-seven studies (27/54) focused on..." or "Five studies (5/27) used decision analytic models...". Please revise throughout the main text.

8) “($98·80-$537·91 per DALY averted)” - please use a consistent presentation format for numbers (i.e. the decimal point). We suggest writing ““($98.80-$537.91 per DALY averted)” - please revise throughout the main text.

9) Please ensure that abbreviations are defined the first time they are used, e.g. 'ICU/intensive care unit', or write the word in full if it is only used a few times. Please revise throughout the manuscript and try to reduce the number of abbreviations used.

10) Please carefully check the tables and make sure that all abbreviations are defined below the table, such as WHO, ICER, QALY, DALY, IU, IM, IV, S/, NHS, GTN, PO, EMTSL.

DISCUSSION

1) Please remove all subheadings within your Discussion, e.g. Implications for policy and practice.

2) Please temper claims of primacy of results by stating, "to our knowledge" or something similar (e.g., “This is, to our knowledge, the first systematic review examining..).

3) “We identified 54 studies - approximately half were conducted in high-income settings.” - if possible, please provide an exact number instead of “approximately half”.

4) Please define abbreviations throughout, such as ‘10IU’.

REFERENCES

1) Please ensure that journal name abbreviations match those found in the National Center for Biotechnology Information (NCBI) databases (http://www.ncbi.nlm.nih.gov/nlmcatalog/journals), and are appropriately formatted and capitalised. For example, in reference [10], should ”International Journal of Gynecology & Obstetrics“ be “Int J Gynaecol Obstet”. Please also see https://journals.plos.org/plosmedicine/s/submission-guidelines#loc-references for further details on reference formatting. 

2) Where website addresses are cited, please specify the date of access (e.g. [accessed: 10/04/2024]).

SUPPLEMENTARY MATERIAL

In the published article, supporting information files are accessed only through a hyperlink attached to the captions. For this reason, you must list captions at the end of your manuscript file. You may include a caption within the supporting information file itself, as long as that caption is also provided in the manuscript file. Do not submit a separate caption file.

a) When SI files are contained with a single file:

Please label the file as ‘S1 Supporting Information’.

Please apply alphabetical labelling to each table and figure contained within the S1 file. For example, ‘Fig A’ to ‘Fig Z’ and ‘Table A’ to ‘Table Z’. Plain text does not need to be labelled and can just be given a title as necessary. For example, ‘Statistical Analysis Plan’. Please cite tables/figures as ‘Fig A in S1 Supporting Information’ and/or ‘Table A in S1 Supporting Information’, for example. Please cite plain text as, ‘Statistical Analysis Plan in S1 Supporting Information’, for example.

b) When SI files are uploaded as separate files:

Please label tables as ‘S1 Table’ (so on) and figures as ‘S1 Fig’ (and so on). Any additional documents (protocols/analysis plans etc.) can be labelled as ‘S1 Protocol’, for example. Please cite items as exactly as labelled.

Comments from Reviewers:

Reviewer #1: Thank you for addrssing comments. one more comment needed to be addressed mentioned in the following: 

- The authors mentioned that they have extracted effectiveness data and compared it to an up-to-date effectiveness estimate from the corresponding WHO recommendation. They identified any effectiveness estimate that differed significantly (i.e., outside of 95% confidence intervals) from an effectiveness estimate found in the literature as having a risk of bias. Could you please clarify in which section you have provided the results for this?

Reviewer #2: I am satisfied with the authors' responses and amendments made to the manuscript. I have nothing further to add. Thank you.

Reviewer #3: The authors have addressed my comments, added to table by region and strengthened the Discussion, I have no further comment.

[LINK]

General Editorial Requests

---

## [Editor Report · Decision Letter 3]

14 Aug 2024

Dear Dr Ginnane, 

On behalf of my colleagues and the Academic Editor, Prof. Andrew Shennan, I am pleased to inform you that we have agreed to publish your manuscript "The cost-effectiveness of preventing, diagnosing and treating postpartum haemorrhage: a systematic review of economic evaluations" (PMEDICINE-D-24-01325R3) in PLOS Medicine.

PRESS

Sincerely, 

Louise Gaynor-Brook, MBBS PhD 

Senior Editor 

PLOS Medicine

lgaynor@plos.org